# Optimizing cell therapy by sorting cells with high extracellular vesicle secretion

Doyeon Koo [1,12], Xiao Cheng[2,3,12], Shreya Udani[1], Sevana Baghdasarian[1], Dashuai Zhu [4], Junlang Li[5], Brian Hall[6], Natalie Tsubamoto[1], Shiqi Hu [4], Jina Ko [7,8], Ke Cheng [4] ✉ & Dino Di Carlo [1,9,10,11] ✉

Critical challenges remain in clinical translation of extracellular vesicle (EV)-based therapeutics due to the absence of methods to enrich cells with high EV secretion. Current cell sorting methods are limited to surface markers that are uncorrelated to EV secretion or therapeutic potential. Here, we utilize a nanovial technology for enrichment of millions of single cells based on EV secretion. This approach is applied to select mesenchymal stem cells (MSCs) with high EV secretion as therapeutic cells for improving treatment. The selected MSCs exhibit distinct transcriptional profiles associated with EV biogenesis and vascular regeneration and maintain high levels of EV secretion after sorting and regrowth. In a mouse model of myocardial infarction, treatment with high-secreting MSCs improves heart functions compared to treatment with low-secreting MSCs. These findings highlight the therapeutic importance of EV secretion in regenerative cell therapies and suggest that selecting cells based on EV secretion could enhance therapeutic efficacy.

Cells produce and secrete numerous bioactive small molecules, proteins, and even larger-scale self-assembled structures comprising multiple biomolecules, such as extracellular vesicles (EVs). EVs play critical roles in intercellular communication and have been increasingly recognized, and are being applied, as therapeutic mediators to drive intracellular signaling in target cells to elicit regenerative and anti-inflammatory properties to treat diseases such as osteoarthritis, pulmonary fibrosis, or myocardial damage[1–7]. In fact, EVs have been hypothesized to be key factors driving powerful therapeutic benefits of mesenchymal stem cells (MSCs) when directly introduced in vivo[8]. However, there are no methods to select cells based on their production and secretion of EVs, and heterogeneity in the propensity of cell therapy products to secrete bioactive compounds like EVs has been highlighted as a potential source for inconsistent therapeutic outcomes[9–15]. Current characterization procedures for EVs and their producing cells use bulk enrichment platforms to isolate secreted EVs from large populations of cells, masking the potential heterogeneity in the secretion phenotype of single cells. No technology has been developed yet to analyze and sort single cells based on EV secretion levels. We hypothesized that technology that can isolate single cells based on high EV secretion could be used to select base cell populations that would be therapeutically superior. Such an approach could also be used to uncover key mediators of biogenesis and secretion pathways for EVs, by combining with transcriptomic assays.

Here, we first developed a nanovial technology to characterize and sort millions of cells based on their secretion of EVs with specific tetraspanin molecular markers (Fig. 1) and then applied this approach to select therapeutic cells for treatment. We found that

[1]Department of Bioengineering, University of California, Los Angeles, Los Angeles, CA 90095, USA. [2]Joint Department of Biomedical Engineering, University of North Carolina at Chapel Hill and North Carolina State University, Chapel Hill, NC 27599, USA. [3]Joint Department of Biomedical Engineering, University of North Carolina at Chapel Hill and North Carolina State University, Raleigh, NC 27607, USA. [4]Department of Biomedical Engineering, Columbia University, New York, NY 10032, USA. [5]Xsome Biotech, Raleigh, NC 27606, USA. [6]Cytek Biosciences, Fremont, CA 94538, USA. [7]Department of Pathology and Laboratory Medicine, University of Pennsylvania, Philadelphia, PA 19104, USA. [8]Department of Bioengineering, University of Pennsylvania, Philadelphia, PA 19104, USA. [9]Jonsson Comprehensive Cancer Center, University of California, Los Angeles, Los Angeles, CA 90095, USA. [10]Department of Mechanical and Aerospace Engineering, University of California, Los Angeles, Los Angeles, CA 90095, USA. [11]California NanoSystems Institute, Los Angeles, CA 90095, USA. [12]These authors contributed equally: Doyeon Koo, Xiao Cheng. ✉e-mail: ke.cheng@columbia.edu; dicarlo@ucla.edu

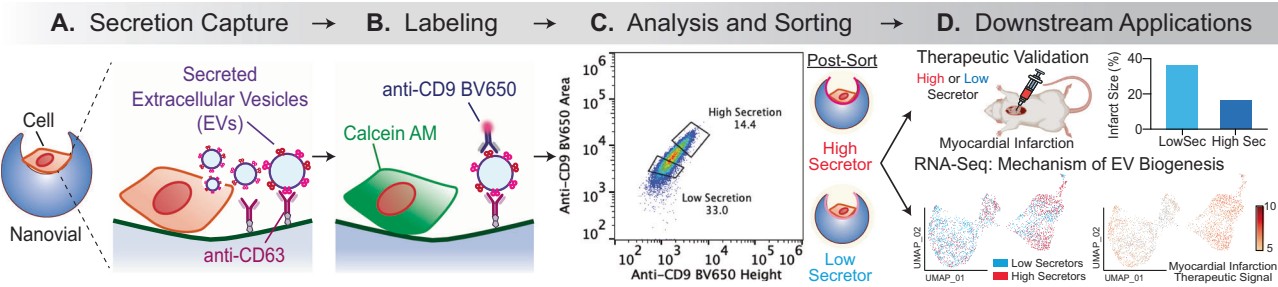

**Fig. 1 | Overview of isolation and downstream analysis of cells based on the amount of EV secretion. A** Cells are loaded in nanovials (blue) that are conjugated with anti-CD63 capture antibodies and allowed to secrete EVs. **B** Secreted EVs from each cell are captured by the surrounding nanovial and labeled with a fluorescent antibody against CD9 on the surface of EVs. **C** Nanovials and corresponding single cells are sorted based on the level of fluorescence signal associated with the captured EVs (High vs. Low Secretors) using a standard cell sorter. **D** Downstream transcriptomic analysis is performed on the High or Low-secretor cells to characterize gene expression differences. In vivo studies are performed with cells expanded from High Secretor or Low-Secretor populations to characterize the effect on therapeutic outcome. **A**, **B**, and **D** created with BioRender.com, and released under a Creative Commons Attribution-NonCommercial-NoDerivs 4.0 International license.

MSCs selected by this phenotype were transcriptionally distinct, expressing markers of EV biogenesis and vascular regeneration, and maintained high levels of EV secretion following sorting and regrowth. We applied the technique to select MSCs based on EV secretion and implanted these cells to treat cardiac injury in a mouse model of myocardial infarction. Animals treated with cells selected based on high EV secretion had improved functional and tissue remodeling outcomes compared to those with low EV secretion. These results support the hypothesis that EV secretion is a therapeutically important aspect for regenerative cell therapies, and can be selected for and/or engineered in the future to improve therapeutic efficacy and reduce batch-to-batch variability.

## Results

### Developing a nanovial assay for EV secretion

Since there are no currently validated surface markers that label high EV-secreting cells, we first had to develop a method to select cells based on their secretion of EVs. Our general strategy made use of cavity-containing hydrogel particles, called nanovials, into which single cells can be seeded. Nanovials were fabricated using a microfluidic device that generates uniform water-in-oil emulsions to create millions of monodisperse polyethylene glycol (PEG)-based nanovials with an inner cavity selectively coated with biotinylated gelatin (Supplementary Fig. 1A)[16–18]. Secreted EVs are captured on the nanovial surface with conjugated antibodies based on expression of unique markers on the secreted EVs (Fig. 1A). The captured EVs can then be labeled with fluorescent antibodies against other surface markers (Fig. 1B). Finally, the live secreting cells can be sorted using standard fluorescence-activated cell sorters based on the level of secretion of EVs for downstream applications (Fig. 1C, D).

### MSC binding and capture of CD63+ CD9+ EVs on nanovials

EVs are loaded with a diverse range of proteins but EVs released from most cell types express membrane-bound tetraspanins such as CD63, CD9, and CD81[19] (Fig. 2A). We determined the combinations of the tetraspanins present on EVs secreted by human immortalized adipose-derived MSCs (iMSCs) using the ExoView platform(Fig. 2B). Ultra-centrifuged EV populations from iMSC conditioned media contained 30% CD63+CD9+CD81+, 37% CD63+, 12% CD9+ and 4% CD81+ EVs. To avoid capture and detection antibodies competing for the same binding sites on EVs and have higher confidence of specific staining of the captured EVs as compared to tetraspanin markers present on the cell body, we targeted CD63+CD9+ EVs using anti-CD63 antibodies to capture EVs and anti-CD9 antibodies for fluorescent labeling of these captured EVs. Strong CD9 signal was observed when EVs isolated from iMSC conditioned medium were incubated with anti-CD63 labeled nanovials, indicating successful capture and detection of CD63+CD9+ EVs (Fig. 2C). There was a two orders of magnitude dynamic range in fluorescence signal measured by flow cytometry for detection of CD63+CD9+ EVs as the concentration of EVs was increased (Supplementary Fig. 1B).

### Analysis and sorting based on single-cell EV secretion using nanovials

Our analysis and sorting are focused on single-loaded cells and their secreted EVs, so we developed cell loading and FACS gating approaches to enrich this population. Optimum single-cell recovery was possible when a ratio of 1 cell per nanovial was used during seeding in a well plate and nanovials with single cells were further enriched based on fluorescence area vs. width signal of a cytoplasmic dye, Cell Tracker Green. Nanovials with single cells had lower fluorescence width signals as compared to those with two cells (doublets) or aggregates of two nanovials with one or more cells (aggregates), showing successful isolation of these populations (Supplementary Fig. 1C). The lower ratio of 1 cell per nanovial instead of 2 cells per nanovial was chosen because it led to the highest fraction of single-cell-loaded nanovials, while limiting the fraction of nanovials with 2 or more cells (<1%).

To characterize the secretion of EVs from cells loaded on nanovials and determine if EV-specific labels could be distinguished from staining of associated cells, we first used imaging flow cytometry (Amnis ImageStream), where staining of both the nanovial surface and cell surface could be independently ascertained. After loading iMSCs onto anti-CD63 labeled nanovials, secreted EVs were accumulated for 24 h and labeled with fluorescent anti-CD9 antibodies. Due to the presence of some staining with anti-CD9 antibodies on the surface of cells, we created a secretion mask on images to isolate the signal coming from captured CD63+CD9+ EVs on the nanovials rather than the signal from cells. The secretion mask comprises a binary mask encompassing the image of the nanovial minus the image of the cell (Fig. 2D). Nanovials gated based on higher overall anti-CD9 fluorescence signal in the secretion mask were observed to have anti-CD9 signal associated with the inner surface of the nanovial cavity, while events with low overall intensity reflected signal associated solely with the cell in the nanovial (Fig. 2E). Notably, the population of nanovials with high CD9-specific staining across the cavity possessed a greater ratio between the area of staining and overall fluorescence intensity than the population with more localized cell staining, mainly due to the spatially distributed secretion signal that spanned the entire surface of the cavity (Fig. 2F). We expected this difference in the distribution of fluorescence intensity across the nanovials could also be detected in non-imaging-based flow cytometers and FACS instruments by using fluorescence peak shape information.

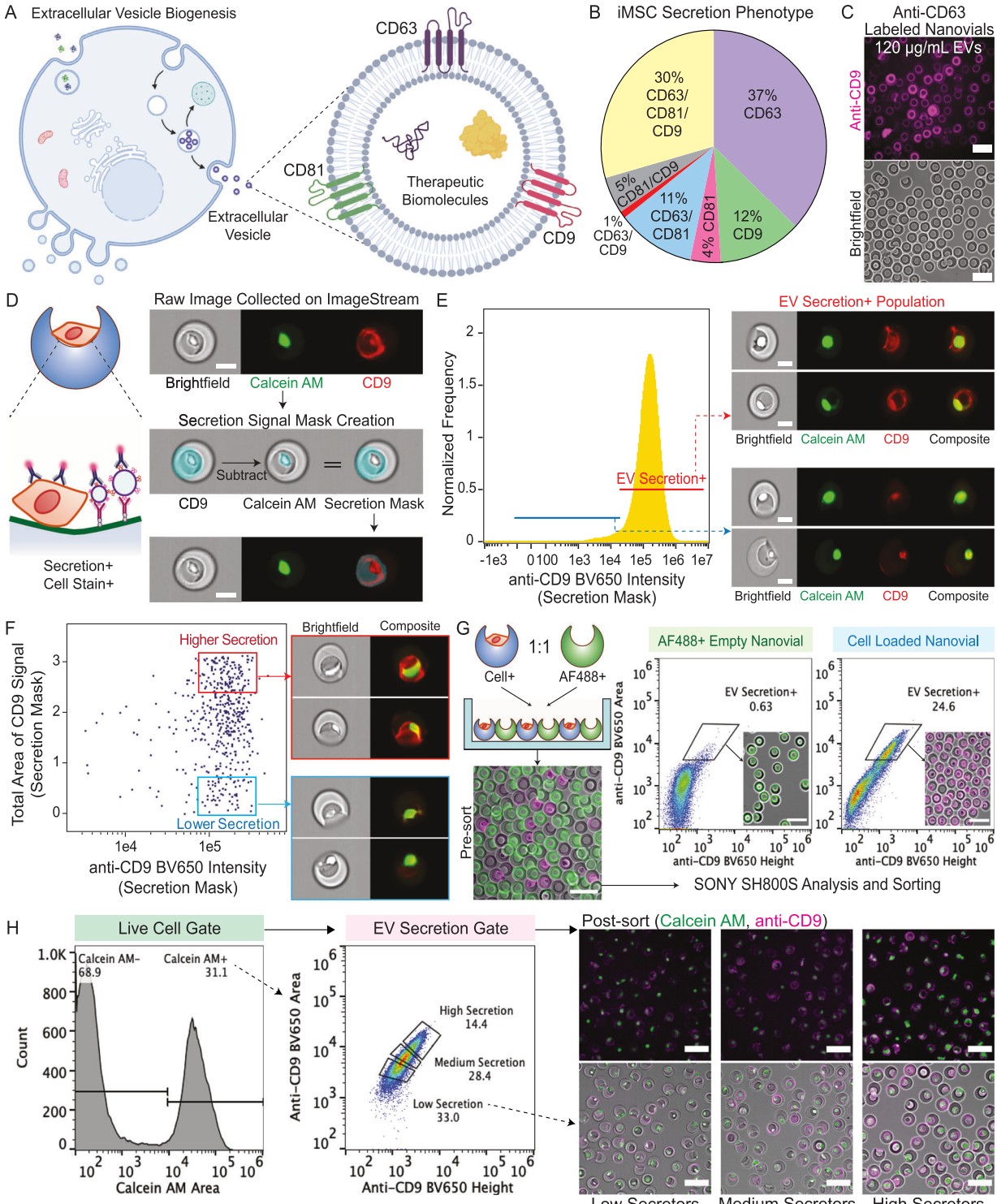

**Fig. 2 | Analysis and sorting of single cells based on EV secretion using nanovials and flow cytometry. A** Schematic of EV biogenesis and expression of tetraspanins (CD63, CD9, CD81) on the surface of EVs. **B** Tetraspanin abundance for EVs secreted by immortalized mesenchymal stem cells (iMSC) analyzed by ExoView. **C** Validation of sandwich immunoassay for detecting secreted EVs on nanovials. EVs were captured on anti-CD63 labeled nanovials and labeled with fluorescent anti-CD9 antibodies as shown in fluorescence and brightfield microscopic images. Scale bars represent 100 μm. **D** Analysis of EV secretion signal on nanovials loaded with cells using ImageStream imaging flow cytometry. A secretion signal mask was created by subtracting the calcein AM signal mask from the CD9 signal mask to exclude non-specific CD9 signals from cell surface staining. Scale bars represent 20 μm. **E** EV secretion+ population represents nanovials loaded with cells and containing high anti-CD9 signals spatially located across the cavity from captured EVs. **F** Differences in the ratio between the area of secretion signal and overall fluorescence intensity between high and low EV secretors result from spatially distributed secretion signals around the entire surface of the cavity. **G** Schematic of an experiment to evaluate crosstalk between cell-loaded and empty nanovials. Less than 1% of empty nanovials were observed to have an EV secretion signal when cell-loaded and empty nanovials were maintained in co-culture for 24 h. Scale bars represent 100 μm. **H** Single cells on nanovials were sorted based on staining for calcein AM and CD9+ EV secretion signal into three categories of high, medium, and low secretors. Fluorescence microscopy images showing sorted populations with corresponding secretion quantity gates. Scale bars represent 100 μm. **A**, **D** created with BioRender.com, released under a Creative Commons Attribution-NonCommercial-NoDerivs 4.0 International license.

Considering the higher fluorescence area vs. overall fluorescence intensity we observed for high EV secretors by imaging flow cytometry, we sorted nanovials based on the fluorescence peak area and height values using a standard cell sorter, the SONY SH800S. When setting a secretion signal threshold from negative control nanovials which were not functionalized with any anti-CD63 capture antibody, we were able to identify a population enriched with CD63+CD9+ EV secretion signal. When sorting events in this gate we found it yielded a mixed population of nanovials containing cells with distributed staining across the nanovial surface, reflecting high and low EV secretors (Supplementary Fig. 2A). We also observed the highest EV secretion signal when secretion was accumulated over 24 h compared to 6 or 12 h of incubation (Supplementary Fig. 2B), providing further evidence that the nanovial signal was associated with EV secretion amount.

Given the long incubation time, we were concerned there could be crosstalk of secreted EVs from cells which could be captured by neighboring nanovials. We evaluated crosstalk between nanovials by co-culturing cell-loaded nanovials and fluorescently labeled test nanovials without cells and found that ~0.6% of test nanovials without cells appeared in the positive secretion gate during the 24-h incubation period, which compared to ~25% of cell-loaded nanovials (Fig. 2G). Presumably, secreted EVs accumulate at higher concentrations locally, and are localized due to their increased size and reduced diffusivity compared to proteins and the shear-protective cavity of the nanovials.

## Isolation and expansion of high and low EV secretors

Having developed the single-cell EV secretion analysis and sorting workflow, our results suggested that there was significant heterogeneity in the secretion of EVs at the single-cell level. From imaging flow cytometry data, we observed a large distribution of secretion signal area, indicating heterogeneity in the number of EVs each cell secretes (Fig. 2F). To better quantify EV secretion at the single-cell level, we simultaneously measured secretion and staining of individual cells with a cell viability dye (calcein AM) (Supplementary Fig. 2C). Across the wide distribution of secreting cells, the majority of cells stained positively with calcein AM indicating viability. >90% of calcein AM positive cells on nanovials secreted EV quantities above the detection threshold for negative control nanovials without capture antibodies. However, a high coefficient of variance (CV = 0.49) in the mean fluorescence intensity (6272) indicated a significant dispersion in secretion among EV secretors.

We next wanted to understand whether the propensity for EV secretion of cells sorted using the assay persisted over multiple generations. To investigate the persistence of selected EV secretion phenotypes, we gated and sorted both single cells and groups of 3000 cells based on EV signal (low, medium, high secretors) thresholding above a negative control sample without capture antibody (Fig. 2H). Recovery of viable cells with different levels of EV secretion was reflected in the fluorescence microscopy images (Fig. 2H). Sorted cells on nanovials expanded in culture actively as cells spread out from the nanovial cavity, adhered to the plate, and divided (Supplementary Fig. 3A, B). Both clonal populations derived from single MSCs and bulk-sorted populations could be expanded over 2 weeks. High secretors from both single-cell and bulk-sorted populations exhibited significantly higher proliferation rates than low secretors (Supplementary Fig. 3C). To account for the higher proliferation rate of the expanded populations, an equal number of cells were seeded after 22 and 13 days of expansion for single-cell colony and bulk-sorted populations respectively, and secreted EVs were quantified from conditioned medium. High secretors were observed to maintain a higher EV secretion rate than low secretors from both single-cell and bulk-sorted colonies, indicating that the EV secretion phenotype was maintained even after multiple cell division cycles (Supplementary Fig. 3D). The EV production rate from single-cell derived colonies exhibited a larger standard deviation in the number of secreted EVs compared to bulk-

sorted populations, suggesting the potential for more measurement error in sorting individual cells or the presence of additional heterogeneity at the single-cell level.

We also wanted to determine if loading MSCs on nanovials affected their multipotency, which could also affect EV secretion. We sorted single-cell-loaded nanovials and expanded and induced differentiation into three lineages using adipogenic, osteogenic, and chondrogenic media. We also sorted iMSCs in culture without loading on nanovials and induced differentiation respectively. Both groups (cells on nanovial vs. original iMSCs) successfully differentiated into the three lineages, showing that growth on nanovials did not affect the multipotency of iMSCs (Supplementary Fig. 3E).

## Transcriptomic analysis of low and high EV-secreting cells

We hypothesized that high EV secretors would have differentially expressed genes compared to low EV secretors and specific molecular mechanisms might drive robust EV shedding. We sorted iMSCs loaded on nanovials as high and low EV secretors based on the CD63+CD9+ EV secretion level and profiled single-cell gene expression for these separate populations using the 10X Genomics Chromium system. We visualized the gene expression of both populations using Uniform Manifold Approximation and Projection (UMAP) and overlaid information on whether each cell belonged to the high and low-secretor populations. Populations representing each secretion phenotype clustered in different regions of the UMAP. High secretor-dominated clusters 1, 2, and 3, while the majority of clusters 4, 5, 6, and 7 contained low secretors (Fig. 3A). iMSC stem cell marker expression was consistent across each cluster, indicating that different sub-populations of iMSCs, not differentiation, may be responsible for the observed heterogeneity (Supplementary Fig. 4A).

To investigate the gene expression profile associated with EV secretion, we first tested whether an EV biogenesis gene ontology signature (GO:0140112 Extracellular Vesicle Biogenesis) marked the clusters with high EV-secreting cells specifically. As expected, the EV biogenesis signal was enriched among high secretor dominant clusters (1 and 2) compared to low-secretor clusters (Fig. 3B). Genes that play a critical role in vesicular trafficking and endosomal recycling processes such as RAS-related GTP binding proteins (RAB7A, RAB27B), tumor susceptibility gene 101 protein (TSG101), hepatocyte growth factor-regulated tyrosine kinase (HGS), and vacuolar protein sorting 4A homolog A (VPS4A) was specifically enriched in these clusters (Supplementary Fig. 4B)[20–22].

As we have observed increased proliferation phenotypes from high EV secretor post-isolation, we further analyzed the expression of genes that positively regulate stem cell proliferation (GO:2000648) (Fig. 3C). All the high secretor dominant clusters (1, 2, and 3) exhibited increased expression of signatures associated with proliferation in stem cells compared to the low-secretor dominant clusters. Some of the upregulated genes in high secretors within this gene ontology were high mobility group A2 (HMGA2), nerve growth factor (NGF), odd-skipped related transcription factor (OSR2), RUNX family transcription factor 2 (RUNX2) and tata binding protein 4 beta (TAF4b) (Supplementary Fig. 4C). HMGA2 enhances stem cell proliferation and reduces aging processes by the increased expression of cyclin E and decreased expression of cyclin-dependent kinase inhibitors[23]. NGF promotes MSC proliferation and chondrogenic differentiation by increasing its sensitivity to surrounding stimulating factors[24].

Since we were interested in applying MSCs in the treatment of cardiac damage, we further analyzed the expression of genes that were previously associated with producing EV molecular cargos known for their therapeutic potential to regulate angiogenesis in treating myocardial infarction[25]. Like EV biogenesis signatures, high secretor dominant clusters 1 and 2 exhibited greater expression of these pro-angiogenic factors (Fig. 3D). These genes included vascular endothelial growth factor (VEGFA), hepatocyte growth factor (HGF), epidermal

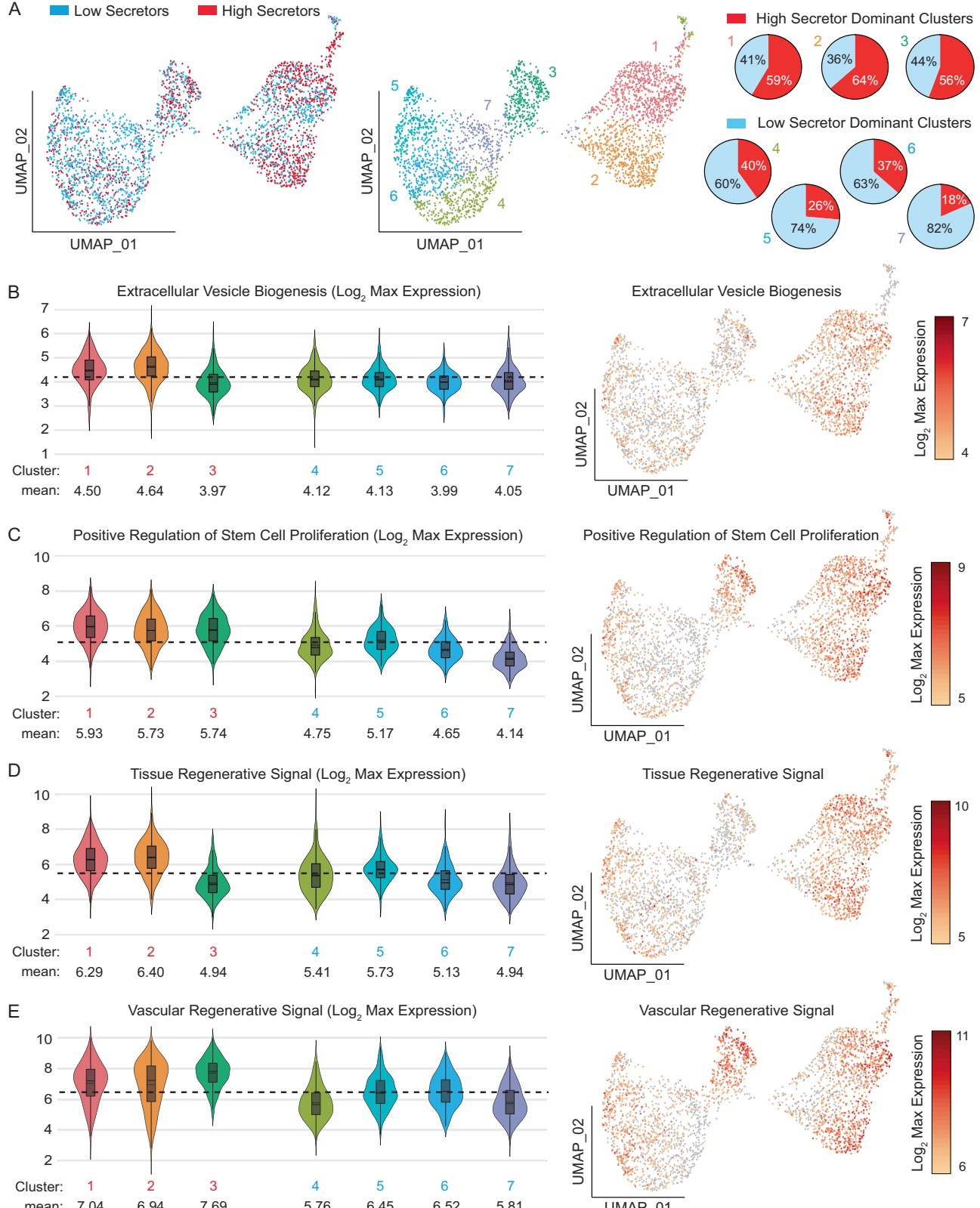

**Fig. 3 | Transcriptomic analysis of low and high EV-secreting cells. A** Uniform Manifold Approximation and Projection (UMAP) representation of high (red) and low (blue) secretors. The distribution of high and low secretors in each cluster is represented as pie charts. Clusters 1, 2, and 3 are high secretor-dominated clusters, while clusters 4, 5, 6, and 7 are low-secretor-dominated clusters. **B** Log₂ max expression level associated EV biogenesis signature (GO:0140112 Extracellular Vesicle Biogenesis) **C** positive regulation of stem cell proliferation (GO:2000648 Positive regulation of stem cell proliferation), and **D** tissue regenerative signal or

**E** vascular regenerative signal represented in each cluster. The Black dashed line represents the average expression level from all clusters. $n = 735$ for Cluster 1, $n = 615$ for Cluster 2, $n = 366$ for Cluster 3, $n = 409$ for Cluster 4, $n = 347$ for Cluster 5, $n = 330$ for Cluster 6, $n = 268$ for Cluster 7. The black boxes in each violin plot represent the first and third quartiles. The black line and black dashed line inside the black box represent the median and mean respectively. The vertical black line extending in the violin plot represents the upper (max) and lower (min) adjacent values.

growth factor receptor (EGFR), nuclear factor kappa B subunit 1 (NKFB1) and CXC-motif chemokine ligands (CXCL1, CXCL2, CXCL3, CXCL6, CXCL8) (Supplementary Fig. 5A). Some of the highly expressed genes are driven by highly expressed transcription factors forming a gene expression node. NFKB1, for example, plays a critical role in driving the expression of other pro-angiogenic genes such as VEGFA and CXCL8[26,27]. Interestingly, only high EV secretor dominant cluster 3 was specifically enriched with another angiogenesis-inducing component, CXC-motif chemokine ligand 12 (CXCL12), which was not expressed in any other clusters (Supplementary Fig. 5A). This result demonstrates that the selection of high EV secretors using nanovials can better identify specific functional base cells expected to promote angiogenesis, and ultimately improve the clinical treatment of myocardial infarction.

We recently also identified another regenerative gene expression signature associated with high VEGFA secretion[28], expected to also be involved in driving angiogenesis, and found the encoding genes for this vascular regenerative signal (VRS) to be more highly expressed in clusters 1, 2, and 3, which were enriched with high secretors. Interestingly, cluster 3, which had reduced EV biogenesis or myocardial infarction regenerative signal mentioned above, showed the highest enrichment in VRS (Fig. 3E). Insulin-like growth factor binding protein 6 (IGFBP6) and heme oxygenase 1 (HMOX1) were the most differentially expressed genes in cluster 3 and fibronectin 1 (FN1) was also overexpressed in the high EV secretor dominant clusters (Supplementary Fig. 5B). Different gene expression among high EV-secreting clusters indicates that there exists transcriptomic heterogeneity even in populations with similar EV secretion levels, suggesting that gene expression of surface marker proteins alone may not be sufficient to mark high EV secretor populations.

Among the top 20 overexpressed transcripts in high secretors, ankyrin repeat and sterile alpha motif domain (ANKS1B), Tenascin XB (TNXB) and platelet endothelial cell adhesion molecule-1 (PECAM1) were the most notable due to their involvement in cell-to-cell signaling, vesicle transportation, extracellular matrix organization, and cell migration (Supplementary Fig. 6A)[29–31]. The top 20 upregulated genes in high secretors were also associated with biological processes related to tissue development, cell migration, and regulation of signaling receptor activity, such as endothelial cell development (GO:0001885), cell-cell adhesion (GO:0098609) and neurotransmitter receptor activity regulation (GO:0099601) (Supplementary Fig. 6B).

## High secretors increase the viability of injured cardiomyocytes via EV-mediated paracrine repair

To evaluate whether the selection of cells based on EV secretion would lead to differences in MSC-mediated tissue repair, we sorted mouse MSCs based on EV secretion levels using nanovials (Supplementary Fig. 7). For screening mouse MSCs, we utilized a similar strategy targeting CD63/CD9 positive EVs using anti-CD63 and anti-CD9 antibodies in our nanovial assay. This double positive population reflected the tetraspanin abundance from mouse MSC-derived EVs analyzed by ExoView (Supplementary Fig. 8A). Scanning electron microscopic images confirmed differences in the amount of captured EVs on the nanovial surface for single-cell-loaded nanovials sorted based on high and low EV secretion signal (Supplementary Fig. 8B). Cell proliferation and EV secretion were both increased in high-secreting (high-sec) mouse cells compared to low-secreting (low-sec) cells as was found for human cells. To compare the paracrine repair ability of these MSCs, we isolated EVs from 1 million cells of each type and incubated them with HL-1 mouse cardiac muscle cells after exposure to hydrogen peroxide. EVs from high-sec MSCs were able to better rescue cells from oxidative stress-mediated apoptosis (TUNEL positive) when compared to the EVs isolated from the same number of low-sec MSCs, which implied an increased paracrine repair activity of high-sec MSCs on a per cell basis (Supplementary Fig. 8C, D).

## High EV secretors have greater therapeutic potential in a mouse myocardial infarction model

To test whether the augmented paracrine activity would translate to increased therapeutic potency in vivo, we performed a head-to-head comparison of the high-sec MSCs to the low-sec MSCs in cardiac repair using a mouse myocardial infarction (MI) model (Fig. 4A). Based on power calculation (power: 0.9, effect size: 0.9, significance level: 0.05, primary endpoint as experimental endpoint Day 28), $n = 7$ per group was sufficient to compare three experimental groups. Before injecting cells into the injured heart, mouse MSCs were suspended in 10 mg/mL hyaluronic acid (HA) hydrogel, acting as a delivery medium to aid cell retention in the heart after injection. As our previous study indicated that intra-pericardial cavity (IPC) injection achieved a higher stem cell engraftment and resulted in a boosted paracrine activity mediated by the stem cell secreted EVs[32–34], we injected high-sec and low-sec MSCs using IPC injection and the same amount of empty HA hydrogel was injected as the control. We monitored changes in the cardiac function one day before, 2 days, 14 days, and 28 days after the surgery (Fig. 4A) by calculating left ventricular ejection fraction (LVEF) and left ventricular fractional shortening (LVFS) from echocardiography (Fig. 4B, Supplementary Table 1). A sharp decrease in LVEF and LVFS at 2 days indicated successful MI in all groups. On Day 14, both LVEF and LVFS started to recover in both high-sec MSC and low-sec MSC treatment groups but not in the control group (Fig. 4C, D). On Day 28, we observed significant differences in the overall left ventricular functions of all experimental groups, with the high-sec MSC treatment group having the highest LVEF and LVFS (Fig. 4C, D). In addition, the cardiac functions of the low-sec MSC treatment group showed a continuous increase throughout the observation period but failed to recover to the baseline level. To further examine the linkage between the EV-secreting ability and therapeutic potential, we referred to our previous study where wild-type MSCs were injected using the same delivery method[32]. Interestingly, the results showed that the 28-day LVEF and LVFS in the wild-type MSC treatment group were intermediate between those of the high- and low-sec MSC treatment groups. This suggests that the therapeutic potential of wild-type MSCs is lower than that of high-sec MSCs, but higher than that of low-sec MSCs, presumably due to the difference in EV-mediated paracrine activity.

The histology analysis of heart sections provided further evidence for the enhanced cardiac repair capability of high-sec MSCs. Specifically, the high-sec MSC treatment group exhibited the smallest infarct size, which was significantly lower than that of both the low-sec MSC treatment group and the control group (Fig. 4E, F). The high-sec MSC also demonstrated the strongest ability to mitigate the cardiac remodeling after MI, which is indicated by the largest left ventricle wall thickness, and the smallest end-diastolic volume (EDV) and end-systolic volume (ESV) (Supplementary Fig. 9A–C). In addition, high-sec MSCs were found to reduce cell apoptosis induced by MI and promote cell proliferation in the heart. This was evidenced by a decrease in caspase-3 expression and an increase in the number of Ki67-positive cells (Fig. 5A, B). Furthermore, an increased density of CD31-positive blood vessels was found in the heart after high-sec MSC treatment (Fig. 5C). Altogether, high-sec MSCs can better mitigate MI injury and help to restore left ventricular function by reducing cell death, promoting cell proliferation, and increasing vascular density around the infarct region.

## High EV-secreting MSC treatment exhibits excellent safety in vivo

Next, we assessed the safety of using high-sec and low-sec MSCs for treating MI. At the experimental endpoint, we harvested major organs (lung, liver, spleen, and kidney) from mice and conducted histology evaluation using hematoxylin and eosin (H&E) staining (Supplementary Fig. 10A). There were no significant differences in the lungs, livers, spleens, and kidneys of the treated mice compared to the healthy

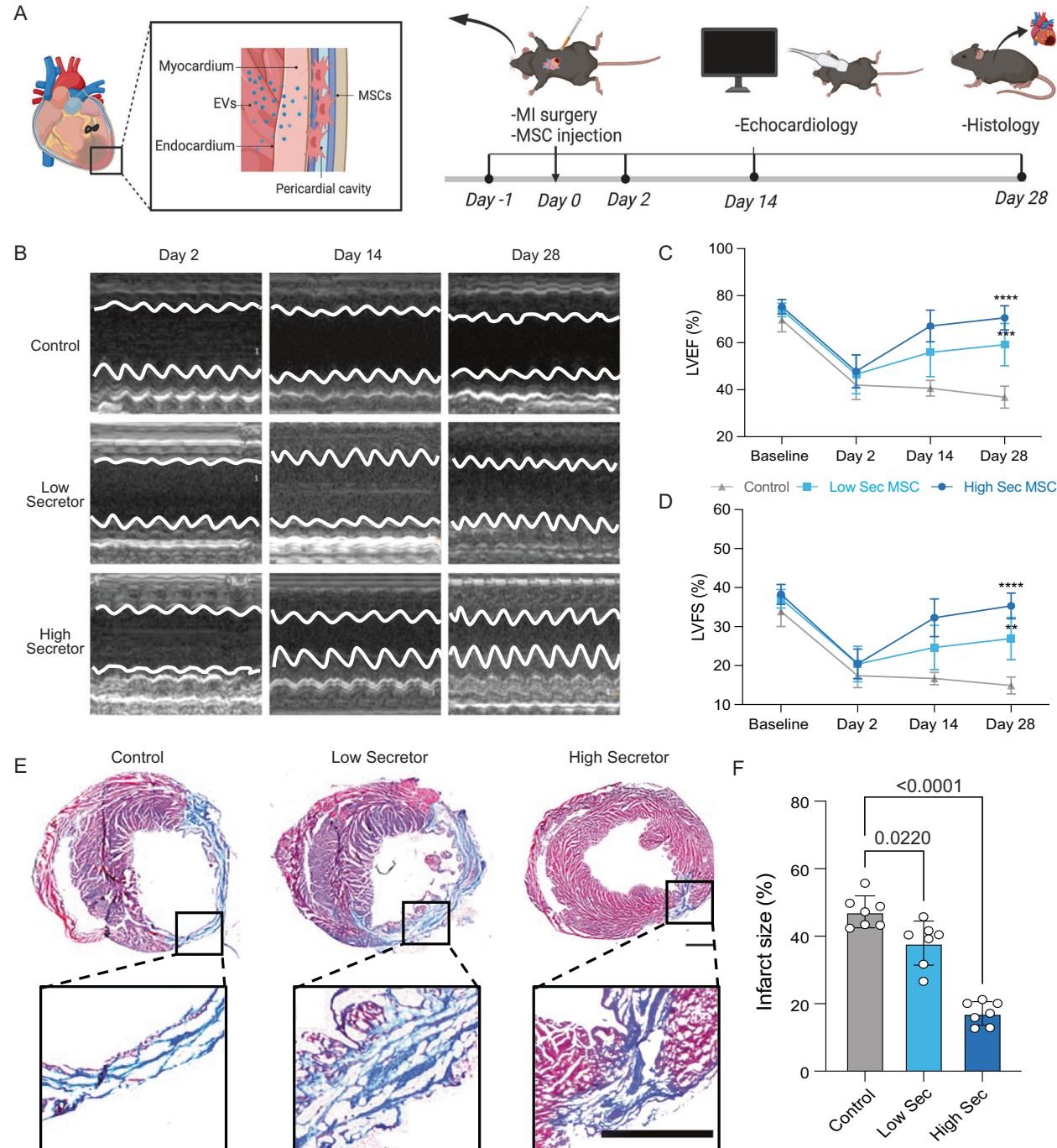

**Fig. 4 | High EV-secreting (high-sec) MSC therapeutic potential compared to low EV-secreting (low-sec) MSCs in a myocardial infarction (MI) model.**
**A** Schematic illustration of the in vivo study design. MSCs were injected into the intra-pericardial cavity (IPC) after MI surgery. Echocardiography was performed at 2 days, 14 days, and 28 days after the surgery. **B** Representative M-mode echocardiography images at 2 days, 14 days, and 28 days post MI surgery. **C** LVEF of mice (control: gray triangle, low EV-secreting MSC: blue square, high-secreting MSC: navy circle) receiving different treatments at 2 days, 14 days, and 28 days post MI surgery (n = 7, biological replicates, the statistical significance is indicated by asterisks above the data points, **$p < 0.01$, ***$p < 0.001$, ****$p < 0.0001$). **D** LVFS of

mice receiving different treatments at 2 days, 14 days, and 28 days post MI surgery (n = 7, biological replicates). **E** Representative Masson's trichrome staining of heart sections (red = healthy tissue; blue = scar) 4 weeks after treatments. The scale bar represents 1 mm. **F** Quantified infarct size of heart sections. All data are means ± SD (n = 7, biological replicates). Comparisons between more than two groups were performed using the one-way analysis of variance (ANOVA), followed by Tukey's honestly significant difference (HSD) post hoc test. The comparisons between samples are indicated by lines, and the statistical significance is indicated by asterisks above the lines. **A** created with BioRender.com, released under a Creative Commons Attribution-NonCommercial-NoDerivs 4.0 International license.

mice, suggesting that high-sec MSC injection does not cause severe side effects such as tumor growth. The blood chemistry results also confirmed that the liver and kidney functions were not affected by high-sec MSC treatment (Supplementary Fig. 10B). Survival curves also demonstrated the safety of high and low EV-secreting MSC treatments

in vivo (Supplementary Fig. 11A). Interestingly, we observed that the high-sec MSC treatment group had lower levels of ALT and AST compared to the low-sec MSC treatment group, which could be due to the increased paracrine activity of high-sec MSCs that helps improve liver functions.

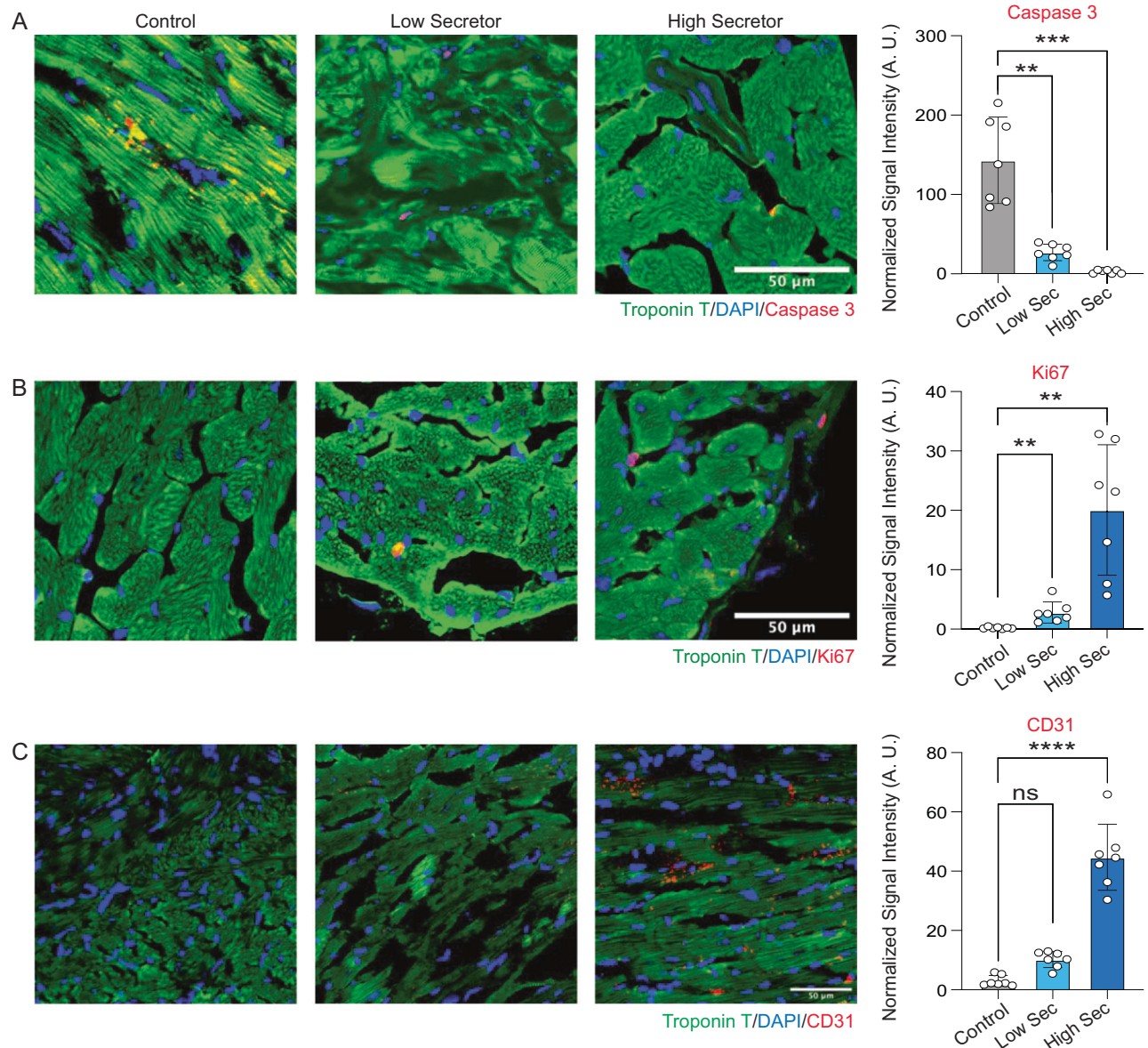

**Fig. 5 | IHC staining of heart sections. A** Representative fluorescence images of cell apoptosis determined by caspase-3 expression (red). The scale bar represents 50 μm. Quantitation of normalized caspase-3 levels in cells ($n = 7$, biological replicates). **B** Representative fluorescence images showing Ki67+ cells (red) in the myocardium. The scale bar represents 50 μm. Quantitation of normalized Ki67 expression in cells ($n = 7$, biological replicates). **C** Representative fluorescence images of vascular regeneration indicated by CD31 staining (red). The scale bar represents 50 μm. Quantitation of normalized CD31 staining in cells ($n = 7$, biological replicates). All data are means ± SD. Comparisons between more than two groups were performed using the one-way analysis of variance (ANOVA), followed by Tukey's honestly significant difference (HSD) post hoc test. The statistical significance is indicated by asterisks above the data points, **$p < 0.01$, ***$p < 0.001$, ****$p < 0.0001$. The comparisons between samples are indicated by lines, and the statistical significance is indicated by asterisks above the lines.

## Discussion

Although there has been increasing interest in using cells that produce EVs therapeutically, previous work did not investigate how selecting sub-populations of cells that secrete more EVs affects therapeutic outcomes. We found that delivering enriched populations of cells that secrete more EVs leads to improved in vitro protection and in vivo functional and structural recovery after a myocardial injury. These results directly link the EV-secretory capacity of cells to therapeutic outcomes and provide further evidence for the role of increased EV quantity in recovery from injury.

Notably, in selecting sub-populations of high EV secretors, we also found that this phenotype was maintained and appeared heritable across multiple cell doublings, indicating a conserved epigenetic program present in a fraction of MSCs that appears to define a more

regenerative sub-type of cells that also proliferates more rapidly. This phenotype is reflected in single-cell transcriptomic measurements which show increased expression of genes previously linked to regeneration and vascular growth, including the vascular regenerative signal (VRS). VRS marks a population of cells that also have high secretion of other factors important for tissue regeneration, like vascular endothelial growth factor A (VEGFA)[18,35]. Given this data, there is a possibility that high EV secretion serves as a marker for a more regenerative and secretory phenotype of cells, and EV quantity itself may not be solely responsible for the therapeutic benefit we observed.

In order to select sub-populations of EV-secreting cells, we needed to develop an assay to measure and sort cells based on EV release at the single-cell level. Multiple studies recently reported immunoaffinity microwell-based platforms for analyzing EV secretion at the single-cell

level, but currently, no technology enables the retrieval of cells[36–38]. Sorting viable intact cells is important for further downstream regrowth or analysis (e.g., single-cell RNA sequencing). Using nanovials we were able to select single cells based on EV secretion level using commercially available instruments and reagents (Sony SH800S 130-micron chip) and directly process cells for transcriptomic analysis using the 10X Genomics single-cell RNA sequencing platform to uncover factors associated with the high EV secretion phenotype. Further studies can quickly build on our results leveraging instruments and nanovials (Partillion Bioscience) that are all available commercially.

Our approach to measure EVs focused on EVs marked with two tetraspanins (CD63 and CD9), however, by adjusting the capture and detection antibodies, different or multiple types of EVs can be captured and analyzed on nanovials. For example, multiple detection antibodies could be applied with different fluorophores, and/or the nanovial surface could be conjugated with more than one capture antibody targeting different EV-surface presented proteins, including engineered targeting protein[39]. We have previously shown the ability to multiplex the detection of secreted cytokines[40]. By further characterizing the types of EVs released, researchers could further subcategorize cells based on the propensity to secrete subsets of EVs and refine our understanding of what are the most important sets of EVs to be released for therapeutic benefit. Although the two tetraspanins chosen for human and mouse cells in our study appeared to reflect overall EV secretion capability, as measured after sorting and re-culture, this may not generalize for other cell sources or species.

The ability to sort cells secreting therapeutically active substances directly can have a large impact on the manufacturing and quality control of EV and cell therapies. Out of 33 registered EV-based clinical trials, 20 clinical trials utilize MSC-derived EVs, where three trials specifically aimed to transplant engineered MSCs for overproduction of EVs[41]. Using nanovials, we can select the most potent cells for growth and delivery, alleviating the challenges associated with significant EV secretion heterogeneity, and batch-to-batch variations in current therapeutics. Given that EV secretion is sustained across multiple cell doublings, we can also select clones for master and working cell banks based on increased EV productivity for direct EV-based therapeutics. Finally, characterizing EV secretion amount and heterogeneity of a finished cell therapy product (e.g., using ImageStream imaging cytometry) may provide a more direct potency assay that ties cell function to therapeutic efficacy, improving the reproducibility and patient outcomes for each therapeutic batch.

## Methods

### Ethical approval
The Institutional Animal Care and Use Committee (IACUC, 22-422) and NIH Guide for the Care and Use of Laboratory Animals were adhered to in all animal research. All studies and protocols were approved by IACUC of North Carolina State University (protocol: 19-811-B).

### Nanovial fabrication
Polyethylene glycol (PEG) biotinylated nanovials with 50 μm diameters were fabricated using a three-inlet flow-focusing microfluidic droplet generator, sterilized and stored at 4 °C in Washing Buffer consisting of Dulbecco's Phosphate Buffered Saline (Thermo Fisher) with 0.05% Pluronic F-127 (Sigma), 1% 1X antibiotic-antimycotic (Thermo Fisher), and 0.5% bovine serum albumin (Sigma) as previously reported[16–18,42]. In brief, PEG pre-polymer, gelatin, and oil phases were infused into a droplet generator microfluidic device and polymerized under UV light through a DAPI filter set on an inverted microscope. Polymerized nanovials were collected in a conical tube and washed to remove unpolymerized phases. A total diameter and inner cavity diameter were calculated using a MATLAB script as reported in our previous study. The inner cavity size for 50 μm nanovials was 30 μm ± 2 μm on

average. As the average size of MSCs is heterogenous (15–30 μm in diameter), the largest diameter of MSCs (30 μm) was considered when choosing the nanovial size (50 μm)[16–18,42].

### Nanovial functionalization
**Streptavidin conjugation to the biotinylated cavity of nanovials.** 50 μm sterile nanovials were diluted in Washing Buffer five times the volume of the nanovials (i.e., 100 μL of nanovial volume was resuspended in 400 μL of Washing Buffer). A diluted nanovial suspension was incubated with an equal volume of 200 μg/mL of streptavidin (Thermo Fisher) for 30 min at room temperature on a tube rotator. Excess streptavidin was washed out three times by pelleting nanovials at 2000 × g for 30 s on a Galaxy MiniStar centrifuge (VWR), removing supernatant, and adding 1 mL of fresh Washing Buffer.

**Anti-CD63 EV capture antibody labeled nanovials.** 50 μm streptavidin-coated nanovials were reconstituted at a five-time dilution in a Washing Buffer containing 270 nM (40 μg/mL) of biotinylated anti-CD63 antibody (Biolegend, 353018). Nanovials were incubated with antibodies for 30 min at room temperature on a rotator and washed three times as described above. Nanovials were resuspended at a five times dilution in a Washing Buffer or culture medium prior to each experiment.

### Cell culture
**Immortalized mesenchymal stem cells (iMSC).** hTERT immortalized adipose-derived MSCs (ATCC, ASC52telo) were cultured based on the manufacturer's protocol using MSC basal media (ATCC, PCS500030) and growth kit (PCS500040). For every secretion assay in this study, cells at passage 18 were used to maintain consistency.

C57BL/6 Mouse Bone Marrow Mesenchymal Stem Cells. Mouse mesenchymal stem cells (cell biologics, C57-6043) were maintained in Iscove's modified Dulbecco's medium (IMDM) (Invitrogen) containing 10% fetal bovine serum (FBS). Cells at passage 8 were used in the in vivo study.

HL-1 cells (Sigma, SCC065) were maintained in Claycomb Medium (Sigma) containing 10% fetal bovine serum (FBS). Cells at passage 6 were used in the in vitro study.

### Nanovial secretion assay general procedure
**Cell loading onto nanovials.** Each well of a 24-well plate was filled with 1 mL of media and 50 μL of 5× reconstituted functionalized nanovials (10 μL of nanovial volume = 100,000 total nanovials) was added in each well using a standard micropipette. Cells were seeded in each well and extra culture medium was added to make a total volume of 1.5 mL. Each well was mixed by simply pipetting 5 times with a 1000 μL pipette set to 1000 μL. The well plate was transferred to an incubator to allow cell binding; the volume in each well was pipetted up and down again 5 times with a 200 μL pipette set to 200 μL at 30-min intervals. Repeated resuspension of nanovials and cells ensured unbound cells that initially did not settle into nanovial cavities in the first loading step could bind to empty nanovials when remixed. Cells that have fallen into a cavity are given 30 min to form initial integrin-based adhesions to the gelatin in the nanovial. After one hour, nanovials were strained using a 20 μm cell strainer to remove any unbound cells and recovered. During this step, any unbound cells were washed through the strainer and only the nanovials (with or without cells loaded) were recovered into a 12-well plate with 2 mL of media by inverting the strainer and flushing with media.

**Secretion accumulation and secondary antibody staining on nanovials.** After cell loading, cells on nanovials were incubated for 24 h in the incubator to accumulate secretion. Each sample was recovered in a conical tube with 5 mL wash buffer and centrifuged for 5 min at 200 × g. The supernatant was removed and nanovials were

reconstituted at a ten-fold dilution in a Washing Buffer containing detection antibody and/or cell viability dye (calcein AM) to label secreted EVs and viability of cells. For each 100,000 nanovials (-10 μL nanovial volume), 5 μL of anti-CD9 BV650 antibody with 85 μL of 0.3 μM Calcein AM solution, unless otherwise stated. Nanovials were incubated with the detection antibody at 37 °C for 30 min, protected from light. After washing nanovials with 5 mL of Washing Buffer, nanovials were resuspended at a 40-fold dilution in the Washing Buffer and transferred to a flow tube.

**Flow cytometer analysis and sorting.** All flow cytometry sorting was performed using the SONY SH800 cell sorter equipped with a 130-micron sorting chip (SONY Biotechnology). The cytometer was configured with violet (405 nm), blue (488 nm), green (561 nm), and red (640 nm) lasers with 450/50 nm, 525/50 nm, 600/60 nm, 665/30 nm, 720/60 nm and 785/60 nm filters. In each analysis, samples were compensated using negative (blank nanovials) and positive controls (purified EV conjugated nanovials labeled with each fluorescent detection antibody or cells stained with calcein AM). Nanovial samples were diluted to approximately 623 nanovial/μL in Washing Buffer for analysis and sorting. Drop delay was configured using standard calibration workflows and single-cell sorting mode was used for all sorting as was previously determined to achieve the highest purity and recovery[43]. A sample pressure of 4 was set for all analysis and cell sorting processes. The following gating strategy was used to identify cells on nanovials with strong secretion signal: (1) nanovial population based on high forward scatter height and side scatter area, (2) calcein AM positive population, (3) CD9 EV secretion signal positive population based on fluorescence peak area and height.

For analyzing samples using the ImageStream imaging-based flow cytometer, a 10 μL nanovial sample was resuspended in 200 μL of Washing Buffer and analyzed based on the manufacturer's protocol. Images were analyzed by IDEAS software distributed by Luminex. Analysis with the ImageStream was repeated two times with three biological replicates for each experiment.

### EV secretion phenotype analysis using ExoView
Human iMSCs were seeded in a T-75 flask at a seeding density of 0.2 million cells per flask with stem cell basal media with 2% FBS. After 24 h of initial seeding, media was changed to OPTI-MEM I reduced serum media without phenol red (Fisher Scientific, 11-058-021). After 48 h, conditioned media was centrifuged at 2500 × g for 10 min at 4 °C and the supernatant was filtered using a 0.2 μm syringe filter (VWR, 28143-310). The sample placed in ultracentrifuge tubes (Fisher Scientific, NC9732446) was placed into a pre-cooled 50 Ti fixed angle titanium rotor and centrifuged in Optima XPN-100 Ultracentrifuge at 120,000 × g for 120 min at 4 °C. Media was removed and the pellet was resuspended in 100 μL cold PBS. The sample was analyzed by the ExoView R100 for CD63, CD9, and CD81 markers based on the manufacturer's protocol. The experiment comprised three biological replicates. For profiling mouse tetraspanin markers, we analyzed EVs concentrated from mouse bone-marrow-derived MSCs using the ExoView mouse tetraspanin kit (CD81 and CD9 capture and CD81, CD63, CD9 as fluorescent counter stains) at the Extracellular Vesicle Core, Children's Hospital Los Angeles.

### Dynamic range of CD63+CD9+ EVs on nanovials
Nanovials were labeled with biotinylated anti-CD63 antibody using the modification steps mentioned above and incubated with EVs isolated from ultracentrifugation at 0, 30, 60, or 120 μg/mL for 2 h at 37 °C. Excess EVs were removed by washing nanovials three times with a Washing Buffer. Nanovials were pelleted at the last wash step and incubated with anti-CD9 as described in the secondary antibody staining procedure above. Following washing three times, nanovials were reconstituted at a 50 times dilution in the Washing Buffer and

transferred to a flow tube. The fluorescent signal on nanovials was analyzed using a cell sorter with sensors and imaged using a fluorescence microscope. This experiment was repeated two times to confirm the reproducibility of EV capture on nanovials.

### Single-MSC loading and statistics
50 μm nanovials labeled with anti-CD63 antibodies were prepared using the procedures described above. To test cell concentration-dependent loading of nanovials, $0.1 \times 10^6$ (1 cell per nanovial) and $0.2 \times 10^6$ (2 cells per nanovial) of cell tracker deep green stained cells were each seeded onto 100,000 nanovials in a 24-well plate, and recovered as described above. Samples resuspended at 40-fold dilution in Washing buffer were analyzed using a SONY sorter for loading efficiency. The population in each gate (single-cell-loaded nanovial, multiple-cell-loaded nanovial, nanovial aggregate) was sorted into 96-well plate containing Washing buffer and imaged using a fluorescence microscope.

### Analysis and sorting of single cells based on CD63+CD9+ EV secretion level
0.1 million MSCs were loaded onto anti-CD63 labeled nanovials and secretion was accumulated for 24 h. The same number of cells were also loaded onto unlabeled nanovials (no capture antibody) as a negative control. After 24 h secreted EVs were labeled with fluorescent anti-CD9 antibodies and calcein AM viability dye. After resuspending nanovials at 40-fold dilution in Washing Buffer, a small fraction of the sample was transferred to a 96-well plate to be imaged using a fluorescence microscope prior to sorting. Samples were analyzed using a cell sorter based on a combination of fluorescence area and height signals. To sort live single cells based on secretion signal, nanovials with calcein AM staining were first gated and high, medium, or low secretors and sorted by thresholding above the negative control sample on CD9 fluorescence area and height signals. Sorted samples were imaged with a fluorescence microscope to validate the enrichment of nanovials based on the amount of secreted EVs captured on the nanovials. This experiment was repeated three times to confirm the reproducibility and isolation of corresponding high and low EV secretors.

### Expansion of cells sorted based on EV secretion level
In each well of a 96-well plate filled with 150 μL stem cell basal media, a single nanovial containing a single cell with a high EV secretion signal was sorted (n = 96). In another two 96-well plates, for each well one nanovial containing single cells with high or medium secretion signal was also sorted. For sorting groups of 3000 cells with the same secretion phenotype, 3000 single-cell-loaded nanovials were sorted based on high, medium, or low secretion gates into each well of a new 96-well plate. Since low secretors proliferated much slower than medium and high secretors, cells were expanded till Day 13 and re-seeded into the T-75 flask at 0.2 million cells per flask. Similarly, the single-cell colony was expanded till Day 22 and re-seeded into T-75 flask at the same seeding density. 24 h after cells were seeded, the media was changed to OPTI-MEM reduced serum media. After 48 h (Day 25 for single-cell colony and Day 16 for bulk-sorted population), conditioned media was collected, and the final number of cells was counted. EVs were collected from conditioned media via ultracentrifugation and quantified by ZetaView analysis at the University of North Carolina Nanomedicines Characterization Core Facility. For calculating EV production efficiency, the total EV count was divided by the total number of cells in each sample.

### Single-cell transcriptomic analysis
The standard protocol for 10X Chromium single-cell 3′ GEX (Chromium Next GEM Single-Cell 3′ Kit v3.1 from 10X Genomics, Catalog number: PN-1000268) was followed unless otherwise noted. Sorted

samples reconstituted at 18 μL were loaded into the 10X Chromium Next GEM Chip for partitioning each nanovial or cell into droplets allowing the PCR amplification and enrichment of gene expression for individual cell barcoded cDNA. Single-cell libraries were constructed using the manufacturer-recommended protocol by the UCLA Technology Center for Genomics & Bioinformatics. Libraries were then sequenced on Novaseq SP (235–400 M/lane) (Illumina). The Cell Ranger pipeline (Cell Ranger Count v7.0.0 with reference (Human (GrCh38) 2020-A) from 10X cloud analysis system) was used for sample de-multiplexing. 10X Loupe browser was used for further analysis and generation of UMAP and violin plots. For EV biogenesis (GO: 0140112) and stem cell proliferation signatures (GO: 2000648), a gene list was downloaded from the Jackson Laboratory. For tissue regeneration and vascular regenerative signatures, a gene list was retrieved from previous studies[25,28].

**In vitro differentiation of iMSC and iMSC loaded into nanovials**
**Cell sorting and expansion.** A human mesenchymal stem cell functional identification kit (R&D systems, SC006) was utilized for functional characterization and comparisons of iMSCs and iMSCs loaded onto nanovials. First, cultured passage 16 iMSCs that had reached 80% confluency were enzymatically harvested with TrypLE (Gibco, 1260413). Half the cells were re-plated for continued culture in a flask while the remaining cells were loaded into nanovials according to the procedures described earlier. Live cells on nanovials were stained with Calcein AM (Invitrogen, C3099) and sorted (Sony Biotechnology, SH800S) into a 24-well plate pre-filled with MSC culture media (8000 events/well). The cells on nanovials were grown in the well plate for a few days until they reached 90% confluency (culture media was changed every 3 days). Upon reaching the desired confluency, the cells were washed once with PBS and detached with TrypLE. The mixture of detached cells and nanovials was then passed through a 37 μm strainer (STEMCELL Technologies, 27215). The nanovials were captured by the strainer and the cells were collected in a 15 ml conical tube, centrifuged, resuspended in MSC media, and counted. In addition, the original cells which were cultured in a flask and grown to confluency (cells that were not loaded into nanovials) were also sorted and expanded in the same manner.

**Induction of adipogenic differentiation and FABP4 staining.** Resuspended cells from nanovials as well as the original cells were seeded into separate wells at a density of $3.7 \times 10^4$ cells/well. The cells were washed into α MEM Basal Media (R&D Systems, CCM007) containing 10% FBS and 1% anti-anti prior to seeding. The plate was incubated for a day in a 37 °C and 5% CO$_2$ incubator. Once the cells reached 100% confluency, the media was replaced with adipogenic differentiation media (α MEM Basal Media containing adipogenic supplement) to induce differentiation. Media in control wells was replaced with fresh α MEM Basal Media only. The media replacement was repeated once every 3 days with freshly prepared media. The culture was continued until day 21 at which point the media was aspirated, cells were washed twice with PBS, and fixed in 4% paraformaldehyde in PBS for 20 min at room temperature. The cells were then washed twice in 1% BSA solution in PBS, and permeabilized and blocked in 0.5 ml of 0.3% Triton X-100, 1% BSA, and 10% normal donkey serum in PBS at room temperature for 45 min. Next, the cells were incubated in 10 μg/mL anti-mFABP4 primary antibody solution overnight at 4 °C. Negative staining wells were also prepared without the primary antibody solution. The next day, cells were washed three times with 1% BSA in PBS and stained with 1:200 diluted NL557-conjugated donkey anti-goat secondary antibody (R&D systems, NL001) in 1% BSA in PBS for 60 min. The cells were then washed three times with 1% BSA in PBS and stained with 1 μg/mL DAPI solution (Thermo Fisher Scientific, 62248) for 4 min. The samples were washed 3 additional times with 1% BSA and placed in PBS for microscopic imaging.

**Induction of osteogenic differentiation and anti-hOsteocalcin staining.** Resuspended cells from nanovials as well as the original cells were seeded into separate wells at a density of $7.4 \times 10^3$ cells/well. The cells were washed into α MEM Basal Media (R&D Systems, CCM007) containing 10% FBS and 1% anti-anti prior to seeding. The plate was incubated for a day in a 37 °C and 5% CO$_2$ incubator. Once the cells reached 70% confluency, the media was replaced with osteogenic differentiation media (α MEM Basal Media containing osteogenic supplement) to induce differentiation. Media in control wells was replaced with fresh α MEM Basal Media only. The media replacement was repeated once every 3 days with freshly prepared media. The culture was continued until day 21 at which point the media was aspirated, cells were washed twice with PBS and fixed in 4% paraformaldehyde in PBS for 20 min at room temperature. The cells were then washed three times with 1% BSA solution in PBS, and permeabilized and blocked in 0.5 ml of 0.3% Triton X-100, 1% BSA, and 10% normal donkey serum in PBS at room temperature for 45 min. Next, the cells were incubated in 10 μg/mL anti-hOstetocalcin primary antibody solution overnight at 4 °C. Negative staining wells were also prepared without the primary antibody solution. The next day, cells were washed three times with 1% BSA in PBS and stained with 1:200 diluted NL557-conjugated donkey anti-mouse secondary antibody (R&D systems, NL007) in 1% BSA in PBS for 60 min. The cells were then washed three times with 1% BSA in PBS and stained with 1 μg/mL DAPI solution (Thermo Fisher Scientific, 62248) for 4 min. The samples were washed 3 additional times with 1% BSA and placed in PBS for microscopic imaging.

**Induction of chondrogenic differentiation and anti-hAggrecan staining.** Resuspended cells from nanovials as well as the original cells were seeded into separate 15 ml conical tubes at a count of $2.5 \times 10^5$ cells/tube. The cells were centrifuged at $200 \times g$ for 5 min at room temperature. The media was removed and replaced with D-MEM/F-12 Basal Media (Gibco, 11320033) containing 1% ITS supplement and 1% anti-anti and centrifuged at $200 \times g$ for 5 min. This procedure was repeated once more. After the last wash, the cells were placed in chondrogenic differentiation media (D-MEM/F-12 Basal Media containing chondrogenic supplement) to induce chondrogenic differentiation. Media in control tubes was replaced with fresh D-MEM/F-12 Basal Media only. The tubes were centrifuged once more at $200 \times g$ for 5 min at room temperature and placed in a 37 °C and 5% CO$_2$ incubator. The media replacement was repeated once every 3 days with freshly prepared media without disturbing the pellet. The culture was continued until day 21 at which point the media was aspirated, each pellet was washed twice with PBS and fixed in 4% paraformaldehyde in PBS for 20 min at room temperature. The pellet was then washed two times with PBS, molded into a cryosectioning mold with Tissue-Tek OCT compound (Sakura, 4583), flash frozen with liquid nitrogen vapor, and stored at −80 °C until cryosectioning was performed. 5 μm-thick sections were obtained using cryostat microtome (Lecia CM1950) and kept frozen until the staining procedure. To stain the pellet sections, the pellet sections were permeabilized and blocked in 0.5 ml of 0.3% Triton X-100, 1% BSA, and 10% normal donkey serum in PBS at room temperature for 45 min. Next, the sections were incubated in 10 μg/mL anti-hAggrecan primary antibody solution overnight at 4 °C. Negative staining sections were also prepared without the primary antibody solution. The next day, the sections were washed three times with 1% BSA in PBS and stained with 1:200 diluted NL557-conjugated donkey anti-goat secondary antibody (R&D systems, NL001) in 1% BSA in PBS for 60 min. The sections were then washed three times with 1% BSA in PBS and stained with 1 μg/mL DAPI solution (Thermo Fisher Scientific, 62248) for 4 min. The samples were washed 3 additional times with 1% BSA, followed by washing once with distilled water. Excess water was removed, and the sections were covered with a drop of mounting medium and a glass coverslip for microscopic imaging.

## Scanning electron microscopy of nanovials

After FACS sorting, nanovials containing both high and low EV-secreting MSCs were subjected to fixation using 4% paraformaldehyde (PFA) (Thermo Fisher, J61899) for a duration of 20 min. Following fixation, the nanovials were thoroughly washed with DI water and subsequently dried at 37 °C on a silicon wafer. Post-drying, a gold coating was applied to the nanovials, enabling the characterization of cell-loaded nanovial morphology through Scanning Electron Microscopy (SEM) (JCM-7000, JEOL).

## In vitro ischemic injury model

To establish the ischemic injury in vitro, HL-1 cells were incubated with 500 μM H2O2 in a complete medium for 2 h. After that, the H2O2-containing medium was replaced by a complete medium with EVs isolated from the same number of MSCs with different secretion abilities. For 24 h treatment, the cell apoptosis was evaluated using TUNEL staining[44].

## IPC injection of MSCs in mice model of MI

The Institutional Animal Care and Use Committee (IACUC, 22-422) and NIH Guide for the Care and Use of Laboratory Animals were adhered to in all animal research. All mice are housed under a 12-h light/12-h dark cycle at a constant temperature of 25 °C, with humidity maintained between 40–60%. To create a mouse MI model, C57BL/6 (Charles River, C57BL/6NCrl, 8 weeks) mice were anesthetized through intraperitoneal injection of K-X cocktail (100 mg/kg ketamine and 10 mg/kg xylazine), while a small animal ventilator (SAR-1000 Small Animal Ventilator, CWE, Inc.) provided artificial ventilation as life support. The LAD coronary artery was ligated permanently with a 6-0 suture under sterile conditions to induce ischemia. Successful induction of Infarction was confirmed by the pale color observed in the apex area. To avoid sex bias, each experimental group comprised 4 male mice and 3 female mice. For the control group, not all mice survived in the first 7 days following the procedure, so this group comprised 12 mice initially. The 5 mice that did not survive by day 7 were excluded from echocardiography and histology studies as they did not make it to the subsequent time points (day 14 and day 28). Immediately after MI induction, 0.2 million cells with 10 μL 10 mg/mL 4% hyaluronic acid (HA) hydrogel were injected into the pericardial cavity of each mouse heart. The chest was then closed, and the animal was permitted to recover.

## Mouse echocardiology

Echocardiography was carried out at various time points, including baseline (prior to MI surgery), and 2, 14, and 28 days after the MI procedure. To conduct transthoracic echocardiography, mice were anesthetized using an isoflurane/oxygen mixture and positioned supine with the body temperature maintained at 37 °C. A 40-MHz probe from the high-frequency ultrasound system (Prospect, S-Sharp, New Taipei City, Taiwan) was used to obtain B (brightness, 2D) and M (motion) mode images. Left ventricular dimensions were measured at both diastole and systole, with five continuous cardiac cycles collected for each animal. The images were recorded and assessed blindly. The ejection fraction was calculated using the following formula: EF = (LVEDV - LVESV/LVEDV) × 100%, while the fractional shortening was determined using FS = (LVEDD - LVESD/LVEDD) × 100%.

## Histology

Mice tissues were harvested and fixed using a neutral buffered 10% formalin solution (NBF, Sigma–Aldrich, St. Louis, MO, USA) overnight. Afterward, the tissues were dehydrated in a 30% sucrose solution at 4 °C for at least one night. Subsequently, the tissues were cryopreserved in optimal cutting temperature (OCT) compound and cryosectioned (CryoStats, Leica). Pathological assessments were carried out using H&E and Masson's trichrome staining in accordance with the manufacturer's instructions (Sigma–Aldrich, St. Louis, MO, USA).

For immunofluorescence staining, cryosections of the tissues were first permeabilized and blocked using a protein block solution (Dako, Carpinteria, CA, USA) containing 0.1% saponin (Sigma–Aldrich, St. Louis, MO, USA). The primary antibodies were then added and incubated overnight at 4 °C, including rabbit anti-Caspase-3 (1:100; ab184787, Abcam, Cambridge, UK), rabbit anti-Ki67 (1:100; ab15580, Abcam), anti-CD31 antibody (1:100; EPR17259, Abcam), and mouse Anti-Cardiac Troponin T antibody (1:100; ab8295, Abcam) to target proteins of interest. Fluorophore-conjugated secondary antibodies were then added for fluorescent imaging. All the tissue slides were mounted by ProLong gold antifade mountant with DAPI (Thermo Fisher, US) before being imaged using an epifluorescent microscope from Olympus. This experiment was performed with $n = 7$ biological replicates for each group.

## Serum chemistry analysis

At the experimental endpoint, blood samples were collected from mice, and serum was isolated using BD Vacutainer® serum collection tubes (BD, 367814). The serum chemistry test was performed by the Department of Clinical Pathology, College of Veterinary Medicine, NC State University.

## Statistics and reproducibility

All experiments were performed at least twice to ensure reproducibility. Analysis and sorting of iMSCs based on EV secretion was repeated three times independently to confirm consistent outputs. We have determined the sample size for the in vivo protocol based on a power calculation (power = 0.9, effect size = 0.9, significance level = 0.05). $n = 7$ per group was large enough to compare three experimental groups. One-way analysis of variance (ANOVA), followed by Tukey's honestly significant difference (HSD) post hoc test was performed with $p < 0.05$, unless otherwise stated. Single-cell RNA sequencing data was analyzed using 10X Genomics Cell Ranger v7.0.0 on 10X Cloud online software, and plots were generated using the 10X Genomics Loupe browser. No statistical method was used to pre-determine sample size and the investigators were not blinded to allocation during experiments and outcome assessment.

## Reporting summary

Further information on research design is available in the Nature Portfolio Reporting Summary linked to this article.

## Data availability

Single-cell RNA sequencing data have been deposited in the Gene Expression Omnibus under accession code GSE240981. Source data are provided with this paper. Because of numerous images and large size (>1 GB), raw imaging data is available upon request. Requests will be filled within 2 weeks. Source data are provided with this paper.

## Code availability

This study did not utilize custom code. Single-cell RNA sequencing data was analyzed using 10X Genomics Cell Ranger Count v7.0.0 with reference (Human GRCh38 20220-A). Further analysis was performed using the 10X Genomics Loupe browser (cloupe) for UMAP generation and differential gene expression.

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

## Acknowledgements

We acknowledge support from the National Institutes of Health grants R21GM142174 to DD and R01HL146153, R01HL144002 grants to K.C. This project has been made possible in part by grant 2023-332386 from the Chan Zuckerberg Initiative Donor Advised Fund (CZI DAF) to D.D., an advised fund of the Silicon Valley Community Foundation. We also acknowledge grant AHA19EIA34660286 from the American Heart Association to K.C. We would like to thank other members of the Di Carlo

lab and Ke Cheng lab for helpful comments and discussion in the preparation of this manuscript. We thank UCLA Jonsson Comprehensive Cancer Center (JCCC) and the Center for AIDS Research Flow Cytometry Core Facility. We thank the UCLA Technology Center for Genomics & Bioinformatics for performing sequencing services. We also thank Robert Knight and Sophie Deng at UCLA for allowing us to use the ExoView human tetraspanin chip and Paolo Neviani at the Extracellular Vesicle Core, Children's Hospital Los Angeles for the use of the ExoView mouse tetraspanin kit.

## Author contributions

D.K., D.D., X.C., and K.C. conceptualized the study and contributed to the design of experiments. J.K. advised iMSC experiments and design. D.K., X.C., S.U., S.B., D.Z., J.L., N.T., and S.H. performed experiments. Formal analysis and visualization of data were conducted by D.K. and X.C. B.H. contributed to analysis of ImageStream data. Funding was acquired by D.D. and K.C. D.K., X.C., D.D., and K.C. drafted the manuscript, and all the authors provided feedback.

## Competing interests

D.D. and the University of California have financial interests in Partillion Bioscience, which commercialized the nanovial technology. The remaining authors declare no competing interests.
