## [Peer Review File · Nature Communications]

Reviewers' Comments:

Reviewer #1:

Remarks to the Author:

The paper addresses the challenge of clinical translation in extracellular vesicle (EV)-based therapies due to the absence of effective methods for enriching single cells with high EV secretion. The authors present a novel nanovial technology that allows the enrichment of single cells based on their EV secretion profiles. This technology was applied to select therapeutic mesenchymal stem cells (MSCs) with high EV secretion, leading to improved treatment outcomes. The selected MSCs exhibited distinct transcriptional profiles related to EV biogenesis and vascular regeneration and maintained elevated EV secretion even after sorting and regrowth. In a myocardial infarction mouse model, treatment with high-secreting MSCs resulted in enhanced heart function compared to low-secreting MSC treatment. This study highlights the therapeutic significance of EV secretion in regenerative cell therapies and proposes that cell selection based on EV secretion could enhance therapeutic effectiveness, which potentially has significant values for regenerative treatments.

I still have a few minor questions regarding this work:

1. In the Results section, specifically the part on "Analysis and sorting based on single-cell EV secretion using nanovials," it is stated that "Optimum single-cell recovery was possible when a ratio of 1 cell per nanovial was used during seeding in a well plate." This conclusion is based on the figure caption of Extended Data Fig.1c, which claims that "The highest fraction of single-cell loaded nanovials was achieved when cells were seeded at 1 cell per nanovial." However, based on the data in the figures, it appears that the fraction of single-cell loaded nanovials for the 1 cell per nanovial group is actually 27.8%, which is lower than that in the 2 cells per nanovial group (40.7%). Can you please clarify this inconsistency?
2. In the second paragraph of the section "Isolation and expansion of high and low EV secretors," it is mentioned that "...we gated and sorted both single cells and groups of 3000 cells based on EV signal..." It would be helpful to provide clarification on what the "single cells" and "groups of 3000 cells" refer to. What are the differences in the sorting conditions for these two groups?
3. In the Materials and Methods section, specifically the part on "Nanovial Functionalization," nanovials with a size of 50 μ m are used for cell capturing. How was this size determined? What is the size of the inner cavity for cells? Is there any characterization result available regarding the correlation between vial/cavity size and cell loading efficiency?
4. In the Materials and Methods section, "Nanovial secretion assay general procedure" part, it is stated that "...the volume in each well was pipetted up and down again 5 times with a 200 μ L pipette set to 200 μ L at 30-minute intervals..." Could you please clarify the purpose of this step? Additionally, in this section, it is mentioned that the "unbound cells were washed through the strainer and only the nanovials (with or without cells loaded) were recovered." Therefore, the nanovials without cells are also kept for future tests. Will this have any effect on the final test results?
5. There are several minor issues that require the author to pay more attention to the details of the manuscript. For example:
 - a) It would be beneficial to include page numbers and even the line numbers in the manuscript to facilitate the review process.
 - b) Fig. 2 uses the uppercase letters for the figure numbers, whereas other figures use lowercase letters. This should be made consistent.
 - c) Fig. 4b may require scale bars for accurate presentation.
 - d) There is an extra space before reference 33 in the sentence "...mode was used for all sorting as was previously determined to achieve the highest purity and recovery 33."
 - e) The sentence "A sample pressure of 4 was targeted" is not clearly presented.

Reviewer #2:

Remarks to the Author:

his paper suggests the possibility to separate the "factory cells" (in this case, human immortalized MSCs) of exosomes that secrete the higher number of exosomes from the low performing (in term of number of exosome secretion) MSCs from the same culture. They also suggest that the MSC which secrete the higher number of exosomes demonstrate very strong cardiac repair and regeneration potential when tested on a mouse model of myocardial infarction. These are very

interesting concepts, but in need of significant refinement and a more detailed explanation of the methods used. The result of the in vivo parts needs scrutiny.

The bioengineering part of the paper is stimulating, but requires significant additional work.

The authors propose to use a new "nanovial" (these vials to incorporate single MSC are indeed larger than the name suggest) system combined with FACS to 1) isolate single cells and 2) separate the single cells that produce more exosomes (here, the system bases the separation on a small subgroup of exosomes, which are double positive for 2 markers, the tetraspanin CD9+ and CD63+) from the single cells that do not do that. Before doing this, the authors have profiled the MSC-exosomes based on the expression of 3 tetraspanins (CD9, CD63 and CD81) in different combination (this was done using the exoview-nanoview technology). The results of the exoview analysis is not in line with the decision to decide to separate the cells based on their capacity to produce high number of double positive CD9 and CD63 exosomes. More tetraspanin combination should have been attempted to 1) confirm that the nanovial system really recognizes the cells with higher production of all the exosome types.

Moreover, different tetraspanins should have been used in parallel, to understand if different tetraspanin profiles are associated with different therapeutic activities of the MSCs.

The Authors should refrain from estimating the secretion of exosomes based on the CD9+/CD63+ combination, only. It is very possible be that the level of CD9+/CD63+ exosomes is not proportional to the total exosome secreted by individual cells.

As an integration and validation of the nanovial/FACS approach, electron microscopy of the nanovial (positive separated and the one not separated, should be shown).

Authors also need to address these questions:

Is the incorporation into nanovial affecting the multi-potency of MSC? Are the MSC stil able to differentiate in the 3 lineages? Authors could perform the differentiation tests.

Are other cell types that MSCs able to to regulate cardiac repair?

More details are needed for these experiments. The Supplement file should include any details of the protocol

Authors have first characterized human MSC-exosomes and then started to use mouse MSCs in cell biology and in vivo mouse work.

Have the authors used the Exoview to assess the tetraspanin profile of mouse MSC? If so, what Nanoview chip have they used?

If specie-specificity needs to be taken into account,

H9C2 cells are derived from embryonic rat heart. It is not logical to use these cells to test the potential of either human or mouse MSC-exosomes. Authors could rather used AC16 Human Cardiomyocyte Cell Line Millipore) or HL-1 (mouse cardiac muscle cell line)

If species-specificity is negligible, why were the in vivo experiments performed using the scarcely characterized mouse MSCs?

The results of the in vivo experiment, especially the electrocardiography showing complete recovery of the cardiac function capacity (LVEF) at 28d after MSC delivery to the pericardial sac are exceptional and need attention/validation. Authors should share the individual electrocardiography data Sample size for the in vivo protocol which is also very small (n=5). How many animals were randomized in the treatment groups? Authors should share their power calculation suggesting to use n=5 mice per group to assess cardiac function(LVEF) by echocardiography.

Do authors have survival curves to show? Has sex as a biological variable been considered? The confocal images of the sections taken from the ischemmic heart are not accurate and need repetition. The heart remodelling data are missing. Other echo parameters have not been shown. The echo machine is not state of the art for this work.

In summary, the paper concepts and technology are stimulating, but some choices are questionable and the in vivo results need scrutiny

Reviewer #1 (Remarks to the Author):

The paper addresses the challenge of clinical translation in extracellular vesicle (EV)-based therapies due to the absence of effective methods for enriching single cells with high EV secretion. The authors present a novel nanovial technology that allows the enrichment of single cells based on their EV secretion profiles. This technology was applied to select therapeutic mesenchymal stem cells (MSCs) with high EV secretion, leading to improved treatment outcomes. The selected MSCs exhibited distinct transcriptional profiles related to EV biogenesis and vascular regeneration and maintained elevated EV secretion even after sorting and regrowth. In a myocardial infarction mouse model, treatment with high-secreting MSCs resulted in enhanced heart function compared to low-secreting MSC treatment. This study highlights the therapeutic significance of EV secretion in regenerative cell therapies and proposes that cell selection based on EV secretion could enhance therapeutic effectiveness, which potentially has significant values for regenerative treatments.

We thank the reviewer for the positive comments about the work and its potential use.

I still have a few minor questions regarding this work:

1. In the Results section, specifically the part on "Analysis and sorting based on single-cell EV secretion using nanovials," it is stated that "Optimum single-cell recovery was possible when a ratio of 1 cell per nanovial was used during seeding in a well plate." This conclusion is based on the figure caption of Extended Data Fig.1c, which claims that "The highest fraction of single-cell loaded nanovials was achieved when cells were seeded at 1 cell per nanovial." However, based on the data in the figures, it appears that the fraction of single-cell loaded nanovials for the 1 cell per nanovial group is actually 27.8%, which is lower than that in the 2 cells per nanovial group (40.7%). Can you please clarify this inconsistency?

We apologize for the confusion for the reviewer as we did not clearly define "Optimum single-cell recovery". We chose a ratio of 1 cell per nanovial instead of 2 cells per nanovial as it provided the highest fraction of single-cell loaded nanovials, while limiting to only a small fraction of nanovials with more than 2 cells in it. We wanted to avoid the presence of multiple cells per nanovial which could skew our understanding of EV production characteristics of single cells. This highlighted our aim of performing secretion-based single-cell analysis by compartmentalizing each individual cell in nanoliter-volume hydrogel microparticles. We have updated the manuscript to describe this rationale for optimizing single-cell loading while reducing multiple cells per nanovial.

2. In the second paragraph of the section "Isolation and expansion of high and low EV secretors," it is mentioned that "...we gated and sorted both single cells and groups of 3000 cells based on EV signal..." It would be helpful to provide clarification on what the

"single cells" and "groups of 3000 cells" refer to. What are the differences in the sorting conditions for these two groups?

We appreciate the request for clarification. For “single cells” in each well of a 96-well plate, we first sorted a single nanovial containing a single cell with a high EV secretion signal. In a second and third plate, we sorted a single-cell loaded nanovial with medium or low secretion signal respectively. Each plate was expanded over 25 days to obtain single-cell derived colonies and we assessed the secretion phenotype of each of these colonies.

Regarding “groups of 3000 cells”, we sorted 3000 single-cell loaded nanovials in each well of a 96-well plate for high, medium, or low secretion gates. Cells in each of the wells were expanded over 16 days to obtain high or low secretor-populations and assessed for maintenance of phenotype. We expanded these cells over a shorter period of time since we started with a larger population of sorted cells initially compared to the “single cells” sort.

We now add some more explanation in the methods section to define this sorting procedure more precisely.

3. In the Materials and Methods section, specifically the part on "Nanovial Functionalization," nanovials with a size of 50 μ m are used for cell capturing. How was this size determined? What is the size of the inner cavity for cells? Is there any characterization result available regarding the correlation between vial/cavity size and cell loading efficiency?

We appreciate the reviewer’s comment. The size of nanovials was analyzed using a MATLAB script as reported in our previous study (de Rutte, ACS Nano, 2022 and Lee, ACS Nano, 2021). In brief, we stained nanovials with fluorescent streptavidin (i.e. streptavidin-AlexaFluor 488) and imaged using a fluorescence microscope. Size distribution characterization was performed using a built-in MATLAB analysis algorithm (i.e. “imfindcircles”). Inner cavity longest dimension (i.e. cavity size) for 50 μ m nanovials was 30 μ m \pm 2 μ m on average. The correlation between outer diameter and cavity size was also reported in our previous study (Lee, ACS Nano, 2021). Generally, the fraction of nanovials with loaded cells increases as the cavity size increases beyond the cell diameter. As the average size of MSCs is heterogenous (15 – 30 μ m in diameter), we tried to match the largest diameter of MSCs (30 μ m) for 50 μ m nanovials. We include more description of these points in the revised manuscript.

4. In the Materials and Methods section, "Nanovial secretion assay general procedure" part, it is stated that "...the volume in each well was pipetted up and down again 5 times with a 200 μ L pipette set to 200 μ L at 30-minute intervals..." Could you please clarify the purpose of this step?

We are happy to provide clarification on the purpose of this step. The purpose of resuspending the nanovial and cell suspension at 30 minute intervals was to ensure unbound cells that didn’t settle into nanovial cavities in the first loading step can bind to

empty nanovials when remixed. Nanovials are heavier than cells, settling faster at the bottom of a well-plate upon resuspension. Due to their orientation, nanovials face up and falling cells can be captured inside the cavity. Cells that have fallen into a cavity are given 30 minutes to form initial integrin-based adhesions to the gelatin in the nanovial and unbound cells that fell on a plate are gently resuspended again to potentially fall into empty nanovials. We add more explanation of the purpose of this step in the revised methods.

Additionally, in this section, it is mentioned that the "unbound cells were washed through the strainer and only the nanovials (with or without cells loaded) were recovered." Therefore, the nanovials without cells are also kept for future tests. Will this have any effect on the final test results?

Nanovials without cells will not affect the secretion capability of other cells inside a neighboring nanovial. However, it will extend our time performing flow cytometer analysis as we will have to screen both empty and cell-loaded nanovials together. One way to avoid this delay is to sort cell-loaded nanovials after cell loading and recovery steps to retain only a cell-loaded nanovial population.

5. There are several minor issues that require the author to pay more attention to the details of the manuscript.

For example:

a) It would be beneficial to include page numbers and even the line numbers in the manuscript to facilitate the review process.

b) Fig. 2 uses the uppercase letters for the figure numbers, whereas other figures use lowercase letters. This should be made consistent.

c) Fig. 4b may require scale bars for accurate presentation.

d) There is an extra space before reference 33 in the sentence "...mode was used for all sorting as was previously determined to achieve the highest purity and recovery 33."

e) The sentence "A sample pressure of 4 was targeted" is not clearly presented.

Thank you for identifying these mistakes. We have edited our manuscript to fix these errors.

Reviewer #2 (Remarks to the Author):

This paper suggests the possibility to separate the “factory cells” (in this case, human immortalized MSCs) of exosomes that secrete the higher number of exosomes from the low performing (in term of number of exosome secretion) MSCs from the same culture. They also suggest that the MSC which secrete the higher number of exosomes demonstrate very strong cardiac repair and regeneration potential when tested on a mouse model of myocardial infarction. These are very interesting concepts, but in need of significant refinement and a more detailed explanation of the methods used. The result of the in vivo parts needs scrutiny.

The bioengineering part of the paper is stimulating but requires significant additional work.

The authors propose to use a new “nanovial” (these vials to incorporate single MSC are indeed larger than the name suggest) system combined with FACS to 1) isolate single cells and 2) separate the single cells that produce more exosomes (here, the system bases the separation on a small subgroup of exosomes, which are double positive for 2 markers, the tetraspanin CD9+ and CD63+) from the single cells that do not do that. Before doing this, the authors have profiled the MSC-exosomes based on the expression of 3 tetraspanins (CD9, CD63 and CD81) in different combination (this was done using the exovoew-nanoview technology). The results of the exoview analysis is not in line with the decision to decide to separate the cells based on their capacity to produce high number of double positive CD9 and CD63 exosomes. More tetraspanin combination should have been attempted to 1) confirm that the nanovial system really recognizes the cells with higher production of all the exosome types. Moreover, different tetraspanin should have been used in parallel, to understand if different tetraspanin profiles are associated with different therapeutic activities of the MSCs.

We appreciate your comment. During our exploratory experiments, we have discovered that the combination of anti-CD63 and anti-CD9 as capture and detection antibody pairs yielded the highest signal on nanovials, while minimizing signal on cells. This reflects what we observed when measuring the immortalized MSC’s EV secretion phenotype as shown with our ExoView Analysis in Figure 2. It will be interesting to explore all other tetraspanin markers in the future. In this study we focused on these two tetraspanins given the performance of the assay and the fact that they are widely accepted as EV markers.

The Authors should refrain from estimating the secretion of exosomes based on the CD9+/CD63+ combination, only. It is very possible be that the level of CD9+/CD63+ exosomes is not proportional to the total exosome secreted by individual cells.

We appreciate the reviewer’s comment. We utilized ExoView analysis to validate our choice of capture and detection antibodies as cells may produce different specific

tetraspanin marker positive EVs. We selected cells based on secretion of EVs with two tetraspanin markers (CD63+CD9+) and from post-growth and secretion analysis, we confirmed that CD63+CD9+ EV secreting cells maintained an overall high secretion phenotype of all EVs. Therefore, we believe the data supports this generalization, although we make sure to emphasize that this may not always be the case.

As an integration and validation of the nanovial/FACS approach, electron microscopy of the nanovial (positive separated and the one not separated, should be shown).

We included scanning electron microscopic images (Supplementary Figure 8B) to show single-cell loaded nanovials and captured EVs on nanovial surfaces after sorting based on high and low EV secretion signal. As shown in the close-up images, high-secreting MSCs have EVs captured near the cavity of the nanovials.

Authors also need to address these questions:

Is the incorporation into nanovial affecting the multi-potency of MSC? Are the MSC still able to differentiate in the 3 lineages? Authors could perform the differentiation tests.

We thank the reviewer for this question. Although we do not believe the multi-potency of the MSCs is relevant for the results on EV secretion presented, we have further tested whether growth on nanovials affects the multi-potency of MSCs. We compared MSC differentiation from the cells on nanovials and the cells that were never loaded onto nanovials. For both populations, we sorted cells and expanded to form a colony and induced differentiation into adipogenic, chondrogenic and osteogenic lineages. We confirmed successful differentiation into the three lineages and that nanovials do not affect the multi-potency of MSCs. This data is presented in new Supplementary Figure 3E.

Are other cell types that MSCs able to regulate cardiac repair?

We thank the reviewer for the question. Apart from the cardiomyocytes, MSCs can also stimulate angiogenesis in the injured heart through EV-mediated paracrine signaling, which is evidenced by the increased CD31 expression in the high EV-secreting MSC treatment group compared to the low EV-secreting MSC treatment group and the control group.

More details are needed for these experiments. The Supplement file should include any details of the protocol

We thank the reviewer for the suggestion, we added more details in the methods section.

Authors have first characterized human MSC-exosomes and then started to use mouse MSCs in cell biology and in vivo mouse work. Have the authors used the Exoview to assess the tetraspanin profile of mouse MSC? If so, what Nanoview chip have they used?

We have additionally performed ExoView analysis to profile tetraspanin markers of mouse MSC-derived EVs. We have discovered that CD63+CD9+ EVs as well as CD63+CD9+CD81+ EVs encompassed 27% of total EVs secreted and therefore was a sufficient marker for secretion of a population of EVs. This new data is presented in Supplementary Figure 8A.

If specie-specificity needs to be taken into account,

H9C2 cells are derived from embryonic rat heart. It is not logical to use these cells to test the potential of either human or mouse MSC-exosomes. Authors could rather used AC16 Human Cardiomyocyte Cell Line Millipore) or HL-1 (mouse cardiac muscle cell line).

If species-specificity is negligible, why were the in vivo experiments performed using the scarcely characterized mouse MSCs?

We thank the reviewer for the suggestion. We have now used HL-1 (mouse cardiac muscle cell line) and repeated the in vitro experiments and confirmed that EVs from high secretors have reduced the cell apoptosis the most. This new data is shown in Supplementary Figure 8C-D.

The results of the in vivo experiment, especially the electrocardiography showing complete recovery of the cardiac function capacity (LVEF) at 28d after MSC delivery to the pericardial sac are exceptional and need attention/validation. Authors should share the individual electrocardiography data. Sample size for the in vivo protocol which is also very small (n=5). How many animals were randomized in the treatment groups? Authors should share their power calculation suggesting to use n=5 mice per group to assess cardiac function (LVEF) by echocardiography.

We thank the reviewer for the comments. We have added mice to this experiment based on the Power calculation below. The effect size is based on expectations of our previous publications (Li, J. et al., Chemical Engineering Journal, 2022., Zhu, D. et al., Nature Communications, 2021.):

Number of Groups: 3

Power: 0.9

Effect size (f): 0.9

Significance Level (α): 0.05

Sample Size: 6.33

Actual sample size: 7

n = 7 per group is large enough to compare three experimental groups. Thus, we performed additional experiments on two mice per group and included those data points in our final analysis. We also improved the quality of the echocardiography data in Figure 4 and included the individual electrocardiography data in the Supplementary Table 1.

Do authors have survival curves to show? Has sex as a biological variable been considered? The confocal images of the sections taken from the ischemic heart are not accurate and need repetition. The heart remodeling data are missing. Other echo parameters have not been shown. The echo machine is not state of the art for this work.

We thank the reviewer for the comments. The survival curve is shown in the Supplementary Figure 11. Each group contains 4 male mice and 3 female mice. We also improved the quality of the IHC images, and the new images are shown in Figure 5. In terms of the cardiac remodeling, we compared the scar thickness, the end-diastolic volume (EDV) and end-systolic volume (ESV) of different groups and the data are shown in Supplementary Figure 9.

Reviewers' Comments:

Reviewer #1:

Remarks to the Author:

The updated manuscript has effectively addressed the questions I raised. I currently do not have any additional inquiries.

Reviewer #2:

Remarks to the Author:

This paper has been reviewed and present improvement. However, the following points still require the attention of the authors.

1. Fig 2B. Could Authors, please, explain how to interpret the content of this panel? This especially for what concerns the minimal proportion (1%) of CD9-positive /CD63-positive (double positive) EVs and the 30% of CD9-positive/CD63- positive/CD81- positive (triple positive). Is the Figure 2B connected in any way with the choice to identify and separate the single cells that produce more exosomes using sorting for CD9 + CD63 This was confusing and it is still confusing after revision.

2. The tetraspanins profile of exosomes secreted from either human (Fig 2B) or mouse (newly provided Suppl Fig 8A) look different.

3. The standard tetraspanin chip of Exoview R100 are designed to the human tetraspanins. Can Authors confirm (within their paper) they obtained mouse tetraspanin chips to be used with mouse MCS-exosomes? Otherwise, they should state (within their paper) that the mouse MSC exosomes can be captured and imaged using the Exoview human tetraspanin chip. The info is important for protocol reproducibility.

4. Fig.S11. Why were the post-MI survival curves performed using different number of subjects? See below, please. Authors have reported the following: "Fig.S11 Survival rate of mice treated with high and low EV secreting MSCs. A) Summary of survival rates of the control group (n=12), low or high EV secreting MSC treatment group (n=7). Low secreting MSC treatment group's survival curve (red) is overlapping with high secreting MSC treatment group curve (blue)".

5. What is the primary end-point used for the power calculation of the in vivo study? This has not been indicated in the revised article.

6. Moreover, the Figure S11 shows that several mice in the negative control group can die before day 14. This mortality is expected, but not in line with the data shown for the echocardiographic analyses.

7. It is incorrect to state that MI and post-MI remodelling is "sex-neutral". It is not. Please, see below (and there is extensive literature on this):

<https://www.nhlbi.nih.gov/health/heart-attack/women#:~:text=Women may get heart attacks,caused by coronary artery disease.>

Reviewer #2 (Remarks to the Author):

This paper has been reviewed and present improvement. However, the following points still require the attention of the authors.

1. Fig 2B. Could Authors, please, explain how to interpret the content of this panel? This especially for what concerns the minimal proportion (1%) of CD9-positive /CD63-positive (double positive) EVs and the 30% of CD9-positive/CD63-positive/CD81- positive (triple positive). Is the Figure 2B connected in any way with the choice to identify and separate the single cells that produce more exosomes using sorting for CD9 + CD63 This was confusing and it is still confusing after revision.

We apologize for the confusion. To clarify, data from Figure 2B was used for validating the heterogeneous populations of secreted EVs and for confirming which tetraspanin markers are present from EVs secreted by immortalized human MSCs. Although the CD9 and CD63 double positive population and CD9, CD63, and CD81 triple positive population encompasses a sizable fraction of the total EVs, other considerations went into choosing CD9 and CD63 for staining as well. This included the specific staining of the captured EVs compared to the cell body. All tetraspanin markers were present on the cell body (CD9, CD63, CD81) but CD9 staining yielded the lowest background signal from the cell membrane. We clarify this in the revised manuscript.

2. The tetraspanins profile of exosomes secreted from either human (Fig 2B) or mouse (newly provided Suppl Fig 8A) look different.

We do not necessarily expect these tetraspanin profiles to look the same. The tetraspanin profiles may look different since the two MSC lines were from different sources. The immortalized human MSCs we used are derived from human adipose tissue, while mouse MSCs used are derived from mouse bone marrow. Other factors, like media for each cell line (MSC basal media for iMSCs and IMDM for mouse MSCs) may have also caused differences in the heterogeneity of secreted EVs.

3. The standard tetraspanin chip of Exoview R100 are designed to the human tetraspanins. Can Authors confirm (within their paper) they obtained mouse tetraspanin chips to be used with mouse MCS-exosomes? Otherwise, they should state (within their paper) that the mouse MSC exosomes can be captured and imaged using the Exoview human tetraspanin chip. The info is important for protocol reproducibility.

We appreciate the reviewer's question. We analyzed mouse MSC conditioned media using a mouse tetraspanin chip (CD81 and CD9 capture and CD81, CD63

and CD9 as fluorescent counterstain) at the Extracellular Vesicle Core from Children's Hospital Los Angeles. We have further clarified this in our methods section.

4. Fig.S11. Why were the post-MI survival curves performed using different number of subjects? See below, please. Authors have reported the following: "Fig.S11 Survival rate of mice treated with high and low EV secreting MSCs. A) Summary of survival rates of the control group (n=12), low or high EV secreting MSC treatment group (n=7). Low secreting MSC treatment group's survival curve (red) is overlapping with high secreting MSC treatment group curve (blue)".

We apologize for the confusion. The control group has n=12 at day 0, with 7/12 surviving at 1 week post the induction of myocardial infarction. The 7 surviving animals were used for the following echocardiography and histology studies. The MSC treatment groups have n=7 from day 0 to day 28 because all animals survived to our end point in these two groups. We had to add additional animals to the control group because we needed all groups to have the same number of surviving animals for the following echocardiography and histology study. We have further clarified this in the figure legend and manuscript.

5. What is the primary end-point used for the power calculation of the in vivo study? This has not been indicated in the revised article.

We apologize for the confusion. The primary end-point for the power calculation is the experimental end point (day 28). We have revised the manuscript to indicate that.

6. Moreover, the Figure S11 shows that several mice in the negative control group can die before day 14. This mortality is expected, but not in line with the data shown for the echocardiographic analyses.

We apologize for the confusion. The dead animals were excluded from the following echocardiography study because they died within 1 week post MI surgery and did not make it to the next time points for echocardiographic analyses (day 14 and day 28).

7. It is incorrect to state that MI and post-MI remodelling is "sex-neutral". It is not. Please, see below (and there is extensive literature on this): <https://www.nhlbi.nih.gov/health/heart-attack/women#:~:text=Women may get heart attacks,caused by coronary artery disease>.

As indicated in our previous response, we have included 4 male mice and 3 female mice for each experimental group to avoid sex bias and clarified this in our methods

section. We double checked and we do not use the term sex-neutral in our currently revised manuscript.

Reviewers' Comments:

Reviewer #2:

Remarks to the Author:

The paper has been fully revised and I do not have additional requests or suggestions for the Authors.